# Spectral Shaping for Neural PDE Surrogates

## Abstract

Neural surrogates for PDE solvers suffer from an inability to model the spectrum of solutions adequately, especially in the medium to high frequency bands. This impacts not only correct spectral shapes, but also stability and long-term rollout accuracy. We identify three convergent factors that exacerbate this phenomenon, namely: distribution shift over unrolls, spectral bias of the MSE loss, and spurious high frequency noise, or *spectral junk*, introduced by the use of pointwise nonlinearities. We find that *spectral shaping*, filtering the spectrum of activations after every layer of pointwise nonlinearities, is enough to reduce spectral junk and improve long-term rollout accuracy. We show spectral shaping not only fixes the learned spectrum (down to machine precision in some cases), but also leads to very stable neural surrogates. We validate these findings on a suite of challenging fluid dynamics problems in the field of neural PDE surrogacy, promoting a clear need for more careful attention to surrogate architecture design and adding a new and simple trick to the practitioner toolbox.

## 1 Introduction

The simulation of physical phenomena is one of the cornerstones of modern science and engineering, enabling the emulation of physics-based systems and the development of digital twin technologies (Ames, 2014; Wagg et al., 2020). Due to the commonplace analytical intractability of physical models, we have seen the development of computational science and numerical methods. Growing interest in building fast surrogate models to partial differential equation solvers, modeling vast areas of the physical sciences, has fueled a boom in the development of neural methods for approximating PDE solutions (Brunton & Kutz, 2023). Anecdotally speaking, however, building such neural surrogates is difficult and, like many other areas of deep learning, is more an art than a science.

For time-dependent PDEs, which we focus on, the standard procedure is to train autoregressive models, such as in Bar-Sinai et al. (2019) or Brandstetter et al. (2022). They require some degree of autoregressive training via backpropagation-through-time (Werbos, 1990), which is slow, memory intensive, and high variance (Metz et al., 2021). Additionally, rollouts are often unstable during test time (Brandstetter et al., 2022) and there are no principled ways to control the error of the rollout. Specifically, there is a lack of rigor in establishing the ideal training setup regarding loss function, architecture, and other tricks for stabilization.

In this paper, we investigate the spectrum of solutions output by such autoregressive surrogate models and demonstrate that correct spectral modeling is important for stable and accurate long-term rollouts. We also note that it is very difficult to model the spectrum correctly with standard neural surrogate baseline architectures. We pinpoint three factors at the root of this poor modeling capability: unroll distribution shift (Brandstetter et al., 2022), spectral bias of the MSE loss (Rahaman et al., 2019), and our key insight, spurious high frequency noise, or *spectral junk*, introduced by the use of pointwise nonlinearities.

Based on these insights, we find out that the simplest remedy, a simple low pass filtering after every nonlinearity, is sufficient to shape the spectrum enough for stable rollouts with increased accuracy. In the rest of this work, we explain our reasoning, with key investigations that connect rollout instability to high frequency signals, and then isolate the source of these high frequency signals as spectral junk introduced by pointwise nonlinearities. We then propose a simple patch, carefully tuned low pass filtering, and demonstrate that this is enough to clean up the spectrum and improve both unroll stability and long-term accuracy. We conduct our experiments with various benchmark-style models on a suite of challenging fluid dynamics problems: Kuramoto-Shivashinsky, Kolmogorov flow, and Rayleigh-Bénard convection, showcasing the efficacy of this one simple trick.

## 2 BACKGROUND

**Partial Differential Equations**  We focus on a class of dynamical systems described by time-dependent partial differential equations. Such systems are ubiquitous in nature. They describe how a solution $u(t, \mathbf{x}) \in \mathbb{R}^d$ evolves in continuous time $t \in \mathbb{R}_{\geq 0}$. These solutions are defined on spatial domain $\mathbf{x} \in \Omega \subseteq \mathbb{R}^n$ beginning from initial conditions $u(0, \mathbf{x}) := u^0(\mathbf{x})$, and subject to boundary conditions such as periodicity or $B[u](t, \mathbf{x}) = 0$ for $\mathbf{x} \in \partial\Omega$, evolves in continuous time $t \in \mathbb{R}_{\geq 0}$. In this work we only consider solutions whose *snapshots* in time $u(t, \cdot)$ are spatially smooth functions. We also only consider autonomous systems of the form

$$\partial_t u(t, \mathbf{x}) = F(\mathbf{x}, u, \partial_\mathbf{x} u, \partial_{\mathbf{xx}} u, ...)|_{u=u(t, \cdot)}. \tag{1}$$

**Discretization**  For computational reasons, *in silico* solutions are discretized in space and time. In space they may be discretized on to a set of points $\mathbb{X} = \{\mathbf{x}_m\} \in \Omega$ with gridded snapshots $\mathbf{u}(t) := \{u(t, \mathbf{x}) | \mathbf{x} \in \mathbb{X}\}$. We may also project against a set of linearly independent basis functions $\mathbb{B} = \{e_1, ..., e_N\}$ obtaining *spectral coefficients* $\hat{u}_n(t) := \langle u(t, \cdot), e_n \rangle$ for appropriately defined inner product and *spectrum* $\hat{\mathbf{u}}(t) := \{\hat{u}_1(t), ..., \hat{u}_N(t)\}$. For example, we would choose a Fourier basis $e_n(x) = e^{-i\pi nx}$ for $\Omega = [-1, 1]$ and periodic boundary conditions, and a Chebyshev polynomial basis $e_n(x) = \cos(k \arccos x)$ for non-periodic boundary conditions.

Similarly, we further discretize in time on to a uniform grid $t_j = j\Delta t$, for $j = 0, 1, ...$ and timestep $\Delta t > 0$, writing $\mathbf{u}^j := \mathbf{u}(t_j)$ and $\hat{\mathbf{u}}^j := \hat{\mathbf{u}}(t_j)$. Successive snapshots are related as

$$\mathbf{u}^{j+1} = \Phi_{\Delta t}(\mathbf{u}^j) \tag{2}$$

where $\Phi_{\Delta t}$, the *flow map*, is found by integrating Equation 1 from $t_j$ to $t_{j+1}$.

**Neural Operator Learning**  The field of neural operator learning (Raissi, 2018; Sirignano & Spiliopoulos, 2018; Rahman et al., 2023; Li et al., 2021; Liu-Schiaffini et al., 2024) is an attempt to learn the flow map $\Phi_{\Delta t}$ associated with a given PDE or family of PDEs, using a neural network $\mathcal{N}$. Neural operators specifically learn to regress the complete snapshot $u$ extending over all $\Omega$, rather than the object $\mathbf{u}$ discretized on to $\mathbb{X}$. For practical purposes, however, it seems that for surrogate models it is acceptable to stick to a fixed resolution.

In this paper, we focus on *autoregressive methods* (Bar-Sinai et al., 2019; Greenfeld et al., 2019; Sanchez-Gonzalez et al., 2020; Brandstetter et al., 2022; List et al., 2024; Schnell & Thuerey, 2024) that map initial conditions $\mathbf{u}^0$ to any future state at time $t_j$ via $j$-times repeated application $\mathcal{N}^j := \mathcal{N} \circ ... \circ \mathcal{N}$. These allow us to roll out to any future point in time, or if $\Delta t$ is fixed, we can at least roll close to it. The problem with autoregressive methods stems in how difficult they are to train. Test time generalization is usually poor when we roll out beyond the longest time unrolled to during training; unrolling at training time is computationally costly; and for chaotic systems it can also be that longer training unrolls provide poorer gradients (Metz et al., 2021; List et al., 2024).

**Stability and Accuracy**  The famous Lax-Richtmeyr equivalence theorem (Lax & Richtmyer, 1956) for classical finite difference solvers states that a solver is *convergent* if and only if it is *stable* and *accurate*. Similar extensions exist for broader classes of solvers (Quarteroni & Quarteroni, 2009, Theorem 1.1). Accuracy refers to polynomial accuracy constraints on a single step; that is, $\|\mathbf{u}(t + \Delta t) - \tilde{\Phi}_{\Delta t}(\mathbf{u}(t))\| = \mathcal{O}(\Delta t^{p+1})$ for small $\Delta t$, integer $p > 0$, and solver flow map $\tilde{\Phi}_{\Delta t}$. Furthermore, the solver must be stable, meaning that given the underlying dynamics are stable, the solver never diverges; that is, $\lim_{N \to \infty} \|\tilde{\Phi}_{T/N}^N(\mathbf{u}(t))\| < \infty$ for all $T > 0$ and $N = \mathbb{Z}_{>0}$. The term $\tilde{\Phi}_{T/N}^N$ refers to the $N$ times repeated application of solver $\tilde{\Phi}_{T/N}$ with timestep $T/N$. These requirements we generally do not satisfy in machine learned systems, but what we lose in guarantees, we gain in speed and ease of deployment.

**Long Term Accuracy and the Distribution Shift Problem**  The major difficulty in training autoregressive models is that good short-term rollout performance does not always translate into good long-term rollout performance. This is due to the *distribution shift problem* (Brandstetter et al., 2022), where one-step errors accumulate over a roll out, shifting the distribution of inputs at timestep $k$ away from the training distribution. This distribution shift pushes the model into a region of data space where it performs poorly. As we shall see, these distribution shifting errors tend to be high frequency signals, emanating from the use of pointwise nonlinearities. Our remedy is to smooth out these errors, projecting model outputs closer to the solution manifold.

## 3 INVESTIGATIONS

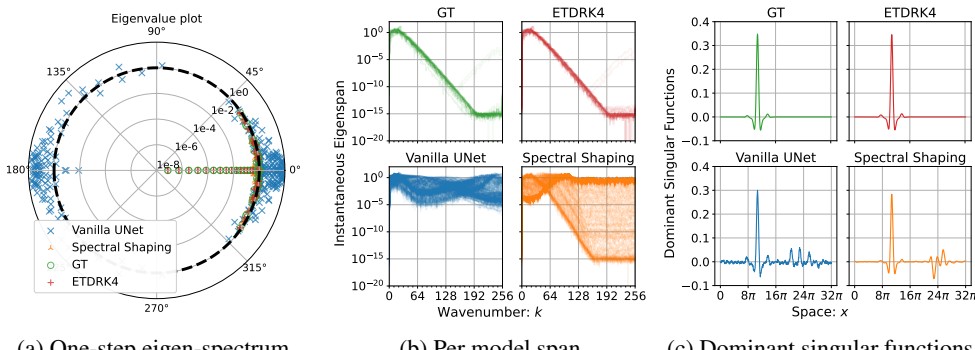

| (a) One-step eigen-spectrum | (b) Per model span | (c) Dominant singular functions |

Figure 1: Eigen-spectra of learned flow maps on the KS equation. (a) Eigenvalue plot for linearized forward models. Inside dashed circle = stable. (b) Instantaneous span of each model. Each shaded line represents an eigenfunction corresponding to an eigenvalue in (a). This represents the total span of the model output, given the current input. We display the log Fourier spectrum of each eigenfunction to highlight the similarities/differences between the spans. All models demonstrate appropriate roll-off apart from the vanilla UNet, which does not. It lacks the capacity to output solutions with the correct spectral profile. Spectral shaping is the key ingredient that pulls the spectrum down into the correct shape in the smoothed model. (c) We also inspect the dominant singular functions of the Jacobian; that is, the singular function corresponding to the largest singular value. All functions look smooth to the eye, except the vanilla UNet, which is rough. Spectral shaping has fixed this.

Below we introduce the phenomenon of spectral junk through a step-by-step investigation into the eigenvalue spectrum of the learned flow $\mathcal{N}$. We target its source—pointwise nonlinearities—and its effect on stability and accuracy of long term rollouts. In the following section we then propose a few methods to mitigate spectral junk, essentially smoothing the spectrum to improve long term behavior. In the results section, we find that improving the spectrum provides an ancillary benefit of improving unroll accuracy and stability.

**Learned flow spectrum** One technique to predict flow in/stability is to analyze the eigenvalue spectrum $\Lambda_{\mathcal{N}}$ of the linearized flow map in the complex plane; that is, the eigenvalues of $\partial_{\mathbf{u}}\mathcal{N}(\mathbf{u})$. For linear, constant-coefficient PDEs the spectrum is constant, but for all other PDEs it is a function of the current solution. Eigenvalues with magnitude greater than unity correspond to directions in solution space that cause instability. Eigenvalues with magnitude less than unity correspond to stable directions. Importantly, the spectrum of the learned flow should match that of the true flow.

Figure 1a compares the spectrum of a learned flow against ground-truth. The ground-truth is obtained by exponentiating the spectrum of the linearized flow as $\exp(\Delta t \cdot \Lambda_F)$. We can also compare this with the linearized flow map of a differentiable off-the-shelf integrator, Exponential Time Differencing fourth-order Runge-Kutta (ETDRK4) (Kassam & Trefethen, 2005). A unit locus is marked by a dashed line. The learned model is green, groundtruth blue, and classical integrator red. We see the underlying flow has many stable eigenvalues on the real axis (close inspection would reveal they are actually complex pairs with extremely small imaginary component), implying convergence in those directions to a low dimensional solution/inertial manifold. Meanwhile, the learned map has many large amplitude eigenvalues, in this case up to $100\times$ larger than unity. The presence of these large eigenvalues explains why sometimes models are seen to rapidly diverge during test time as documented in Brandstetter et al. (2022). Probing these eigenvalues, we see that most of them are high frequency modes (see Figure 2 for visualization). This begs a number of questions: 1) Why is the model unstable in the very the directions the ground-truth flow map assigns as stable?; 2) Does it matter that the learned flow is unstable in these high frequency directions, if we perhaps are unlikely to visit them?; and 3) Why are error vectors typically high frequency?

**1) Stable $\mapsto$ unstable** Is it a coincidence that stable directions are mapped to unstable ones? We argue no, in fact, without special attention, stable directions will generally be very difficult to learn. Since we gather training examples from the ground-truth (or close to the ground-truth) solution, the one-step model only ever sees inputs and targets that are trapped close to the solution manifold—

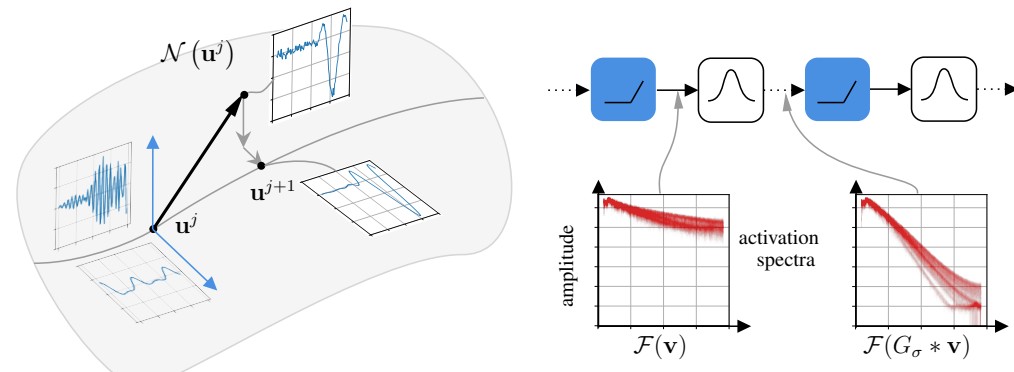

Figure 2: LEFT: Decomposition of predictive error into in-solution manifold and out-of-solution manifold components. The schematic shows a snapshot $\mathbf{u}^j$ mapped by a neural model to point $\mathcal{N}(\mathbf{u}^j)$. This deviates from the ground truth $\mathbf{u}^{j+1}$ in two ways, an in-manifold error (diagonal gray arrow) and an out-of-manifold error (vertical gray arrow), where the manifold in question is the solution manifold of the given PDE. The blue arrows represent in/out-of-manifold directions. Out-of-manifold directions tend to be of higher frequency, demonstrated by the high frequency eigenfunction, while in-manifold directions tend to be smoother, as shown by the low frequency eigenfunction. Much of the noise, pushing the solution $\mathcal{N}(\mathbf{u}^j)$ off the solution manifold, originates from the use of pointwise nonlinearities, which inject *spectral junk* and distort the frequency content of solutions. RIGHT: Spectral shaping addresses spectral junk by applying a smoothing filter, effectively projecting $\mathcal{N}(\mathbf{u}^j)$ back down, or at least closer, to the solution manifold. This mitigates the distribution shift problem (Brandstetter et al., 2022) and improves stability and long-term unroll accuracy.

smooth solutions with little high frequency content. Indeed, it appears that small eigenvalues of the linearized model are hard to learn precisely because they are small. Being small, they suppress signals in that direction of solution space, and thus we do not typically see solutions of that form in the training set, leading to poor performance.

**2) Are unstable directions a problem?** Although the one-step model has regions of its input space that are unstable, one might argue that if such an input is highly unlikely, by virtue of not being in the training set, then we need not learn to model it properly. The problem with this line of thinking is that the one-step output is generally not on the solution manifold but perturbed off it. Figure 3a shows where the UNet outputs solutions with mid-to-high frequencies up to $10^{10}\times$ larger than observed. The error vector from the one-step model points precisely into the unstable region. This is an example of the so-called *distribution shift problem* (Brandstetter et al., 2022) where the distribution of inputs to the one-step model at unroll step 2 is shifted from the training set.

**3) Why are the unstable directions high frequency?** It appears that the error vectors output by learned models contain significant high frequency content (c.f. Figure 3a), which we call *spectral junk*. We pinpoint two mechanisms for this: one architectural (pointwise nonlinearities) and the other loss-based (spectral bias (Rahaman et al., 2019)). Pointwise nonlinearities inject high frequency noise into the spectrum and also lead to high amounts of aliasing (McCabe et al., 2023; Raonic et al., 2024; Karras et al., 2021). The right hand pane of Figure 3b shows the effect of different activation functions on a sample snapshot from the Kuramoto-Shivashinsky training set. All activation functions considered introduce severe uplift in the high frequency band, between 7 to 12 orders of magnitude. A single layer of length 3 convolutional filters is not enough to smooth out these high-frequencies while maintaining the low order modes. Secondly, typical one-step training with the MSE loss, without any modifications, preferentially learns low frequencies over high frequencies (Rahaman et al., 2019). This is a well-known phenomenon, called *spectral bias* of neural networks. A more accurate description would be to term this spectral bias of the MSE loss, or low-pass filtering from a signal processing viewpoint. Low frequencies dominate the loss, a condition

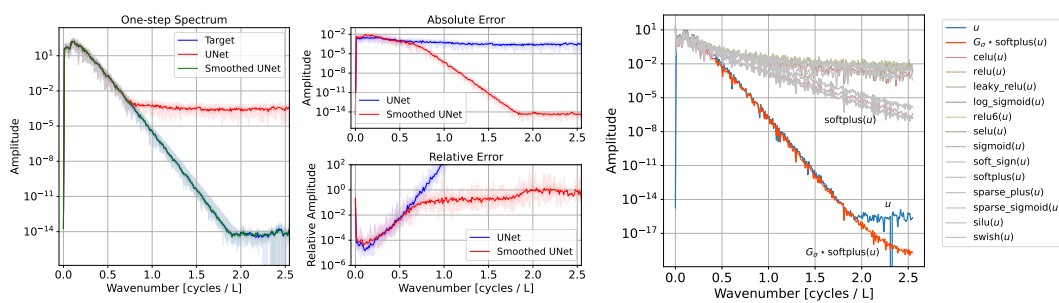

(a) Fourier spectrum on the KS equation   (b) Action of nonlinearities on spectrum

Figure 3: (a) Fourier spectrum for one-step snapshots on the KS equation. The bold lines represent the median power spectral density averaged 16 random snapshots, with shading representing the min-max spectra. Note the `float64` round-off error around amplitude `1e-14`. This is limit of accuracy. Notice the drastic improvement in the spectrum of the smoothed model, which overlaps the true spectrum down to the `float64` precision floor. (b) Effect of various operations on the spectrum of a snapshot from the 1D Kuramoto-Shivashinsky equation. All nonlinearities (in muted colors), introduce heavy distortion into the high frequency components. The softplus produces the smoothest spectrum, which we further improve by applying the discrete Gaussian filter, yielding $G_\sigma \star \text{softplus}(u)$, which has an almost identical spectrum to the underlying signal.

guaranteed for smooth signals[1], and so most of the optimization effort is placed in learning these dominating frequencies. Achieving an MSE loss of say `1e-4` or lower on high frequencies is useless for one-step training, and so from an optimization perspective we tolerate such errors, but if we consider the unrolled system high frequencies are indeed important.

**Related work on stabilization strategies** *Autoregressive training*, also known as Backpropagation Through Time (BPTT) (Werbos, 1990), is the simplest training technique. During training, the network is unrolled for as many timesteps as there are in the training sequence, with the time and memory complexity scaling linearly with sequence length. For increased efficiency, it is common to truncate trajectories into shorter snippets (e.g., (Brandstetter et al., 2022)). But unroll too little and the model still goes unstable, unroll too much and the signal-to-noise ratio of backpropagated errors is too small and slows down learning (List et al., 2024). DySLIM (Schiff et al., 2024) is a recently proposed approach, which is autoregressive in nature, but uses a different loss. It minimizes the measure-theoretic distance/divergence of a bundle of prediction trajectories to the training data. One angle on DySLIM is that it tackles the distribution shift problem directly. Another is that it circumvents spectral bias, by switching to a different loss function.

There is also a broad class of *projection-based* techniques: *Denoising* corrupts the input to the network such that we minimize the loss $\mathbb{E}_\epsilon L(\mathbf{u}(t_1), f(\mathbf{u}(t_0) + \epsilon))$. Sanchez-Gonzalez et al. (2020) choose Gaussian $\epsilon$ and later Mayr et al. (2023) choose Brownian motion noise for $\epsilon$. This noise mimics the high frequency perturbations off the solution manifold, unseen in one-step training but introduced in rollouts. Choice of this noise distribution is empirical and requires hand-tuning and so the *pushforward trick* (Brandstetter et al., 2022), a special case of Truncated BPTT (e.g., (Williams & Zipser, 1995)) was introduced to sample $\epsilon$ directly from the *distribution of errors*. For this, a model is unrolled 2 steps in training, but gradients are only backpropagated through 1 step. It is a simple and effective trick. Lippe et al. (2023) takes the one-step denoising step to the extreme with PDE-Refiner, designing an iterative denoising procedure. They learn two kinds of map. A one-step map to push a solution one step forward in time $\mathbf{u}(t_0) \mapsto \hat{\mathbf{u}}^1(t_1)$, where $\hat{\mathbf{u}}^1(t_1)$ is a first candidate for $\mathbf{u}(t_1)$. They then learn a series of denoising maps, which successively refine $\hat{\mathbf{u}}^1(t_1) \mapsto \hat{\mathbf{u}}^2(t_1) \mapsto \cdots \mapsto \hat{\mathbf{u}}^J(t_1)$, for $J$ steps. Each successive denoising map is trained as a denoising auto-encoder on a exponentially decreasing noise schedule. While PDE-refiner is effective, it requires multiple iterations per time step during inference. It is also not clear how PDE-Refiner fares when combined with autoregressive training, since this was never shown in the original paper.

---

[1]For smooth signals, $f(x)$, we know the Fourier transform of the $n^{\text{th}}$ derivative decays at least as fast as $|k|^{-n}$, for wavenumber $k$

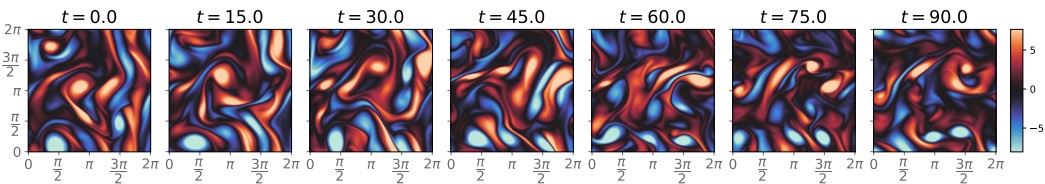

Figure 4: Example trajectory for the vorticity field from the 2D Kolmogorov flow equation at Re = 500. The flow is defined on a periodic domain exhibiting isotropic spectrum and stationarity.

All the above methods work to stabilize unrolled dynamics. Sometimes they can be combined (e.g. denoising and autoregressive training). The effectiveness of each method varies per system. In this paper, we introduce a new, simple technique for improving the spectrum that can can also aid stabilization. It too can be combined with the aforementioned techniques. Crucially, it adds a new operation that remedies poor spectral modeling, important for modeling of PDEs.

We observe for all the systems we consider, the perturbations are all high frequency. Therefore, instead of learning a projection operator back on to the solution manifold, we can directly build one by filtering out the noise. This turns out to be both remarkably simple, but also cheap and effective. Furthermore, we identify the pointwise nonlinearity as the source of spurious high frequency. Placing a simple blurring operation after each pointwise nonlinearity, we can improve the spectrum, stabilize roll-outs, improve rollout decorrelation time, and reduce the need for autoregressive training.

## 4 METHOD

The method is straightforward. We simply low pass filter after *every* nonlinearity. After trying some alternatives, we find that convolution with a discrete Gaussian filter (Lindeberg, 1990) works well. It is simple and effective. The discrete Gaussian $G_\sigma$ has the form

$$G_\sigma(n, \sigma^2) = e^{-\sigma^2} I_n(\sigma^2) \overset{\text{Fourier}}{\Longleftrightarrow} \hat{G}_\sigma(k, \sigma^2) = e^{\sigma^2(\cos\frac{2\pi k}{N} - 1)} \tag{3}$$

where $I_n$ is the modified Bessel function of integer order $n$, $n \in \mathbb{Z}$ being 1D pixel coordinates. We include the Fourier transform $\hat{G}_\sigma$ with frequency $k$ measured in cycles per $N$ pixels, which is more efficient to implement for large $\sigma$. We do not notice any slow down. For clarity, if before we used nonlinearities $\nu$, we now replace *every* $\nu$ with

$$\mathbf{v} = \nu(\mathbf{u}) \implies \mathbf{v} = G_\sigma \star \nu(\mathbf{u}), \tag{4}$$

for pre-activations $\mathbf{u}$ and post-activations $\mathbf{v}$. We find that the choice of $\sigma$ is important. Too much and the network is too capacity constrained to learn any useful features. Too little and we do not filter out all the high frequency spectral junk. We select it with a hyperparameter sweep, finding that values in the range $\sigma \in [1, 10]$ tend to work well across all systems we experiment on. For 2D filtering, we simply filter each dimension independently, tying Gaussian bandwidths when we expect isotropic spectra.

## 5 EXPERIMENTS

**Systems** We demonstrate the utility of spectral shaping on a range of PDE benchmarks. In 1D, we focus on the Kuramoto-Shivashinsky (KS) equation, a fourth-order chaotic 1D equation, on a periodic domain. The power spectral density exhibits strong roll-off, spanning 16 orders of magnitude, as exhibited in Figures 1a & 3b. In 2D we focus on Kolmogorov flow (KF) as per Kochkov et al. (2021b), defined on a 2D periodic domain with velocity field $\mathbf{u}$. We choose to model at Reynolds number Re = 500, which exhibits strong roll-off in the Fourier domain. We also introduce Rayleigh-Bénard convection (RBC), a model of turbulent thermal convection, between two plates, heated below and cooled from above. It contains non-periodic boundary conditions at the two plates, turbulent decay, non-stationarity, and extreme sensitivity to initial conditions. Complete descriptions of the equations and how training/test data are generated can be found in Appendix A.

**Baselines** We focus on the UNet (Ronneberger et al., 2015) as a baseline. Gupta & Brandstetter (2022) and Stachenfeld et al. (2022) found this to be a strong model for capturing multiscale phenomena. For exact architectural details, check out Appendix B. We train using the standard MSE

objective. For unroll stabilization, we experiment with initial condition noise injection (denoising) (Stachenfeld et al., 2022), the pushforward trick (Brandstetter et al., 2022), autoregressive training on truncated trajectories, and, of course, spectral shaping. Note, we only consider deterministic baselines in this work. For all methods, we sweep the defining hyperparameter of interest (e.g., Gaussian width) and select the best performing model via cross-validation across training runs, as one would usually do in a practical setting.

**Evaluation metrics**   We characterize spectral unroll accuracy in the near- and far-term regimes with an assortment of metrics that characterize the gap between predictive and groundtruth spectra. In the far-term, we use the *mean energy log ratio* (Wan et al., 2024)

$$\text{MELR} = \frac{1}{|K|} \sum_{k \in K} \log E_{\text{pred}}(k)/E_{\text{true}}(k), \qquad E(k) = \sum_{\mathbf{k} \in S_k} \frac{1}{2}|\hat{\mathbf{u}}(\mathbf{k})|^2 \tag{5}$$

for one-sided 1D energy spectrum $E(k)$ with scalar wavenumbers in $K$, and averaging sets $S_k$. For instance, for radially averaged spectra, we use $S_k = \{\mathbf{k} \in \mathbb{R}^2 \mid |\mathbf{k}| = k\}$. The MELR measures the integrated gap between the log energy spectra of the predicted and true solutions at a single point in time. For 1D (KS) and isotropic turbulence (KF) we use radially averaged spectra; for RBC we average parallel and perpendicular to the plates. For exact details, see Appendix D. To compute spectra we use the real DFT in the periodic direction and the type-II DCT (Makhoul, 1980) in non-periodic directions. Note the MELR is invariant under multiplicative rescalings of the energy of the form $c(k)E(k)$ for any nonzero function $c$ of wavenumber $k$.

Short term accuracy is measured with *decorrelation time* $\tau_c \geq 0$, the shortest time until which the *correlation* $C(\mathbf{u}(t), \mathbf{u}_{\text{GT}}(t))$ between prediction $\mathbf{u}(t)$ and groundtruth $\mathbf{u}_{\text{GT}}(t)$ crosses below threshold $-1 \leq c \leq 1$. We mainly report $\tau_{0.8}$. For Kolmogorov flow, we instead measure the *vorticity correlation* $C(\omega(t), \omega_{\text{GT}}(t))$ for vorticity $\omega = \partial_x u_2 - \partial_y u_1$, as per Kochkov et al. (2021b). We implement the directional derivative operators using spectral differentiation (Trefethen, 1996, Ch. 7,8).

## 5.1 RESULTS

**Manifold Accuracy**   Phase space diagrams serve as one tool to sanity check the solution manifolds of dynamical systems. Following Gao et al. (2024, Fig. 3), we plot a 2D histogram of $\mathbf{u}_x$ and $\mathbf{u}_{xxx}$, averaged over time and space. In Figure 5, we have lined up the phase space diagrams for all tested systems on the KS equation. Denoising and spectral shaping all perform well; however, spectral shaping has the lowest two-sample Kolmogorov-Smirnov distance to groundtruth. Plots visualizing combinations of derivatives up to 5[th] order can be found in Appendix C.1, where we show that spectral shaping captures the solution manifold well.

**Raw performance of spectral shaping**   Figure 6(a) shows the impact of spectral shaping on MELR and decorrelation times $\tau_{0.8}$ & $\tau_{0.9}$ in combinations with all equations and baselines. We measured the spectral gap between predictions and ground truth 128 & 256 timesteps out from the initial conditions. Spectral gap is measured in terms of mean energy log ratio (MELR). For RBC we only report the perpendicular velocity spectrum, since the other three (parallel velocity, perpendicular buoyancy, and parallel buoyancy) tell the same story. Those results are listed in full in

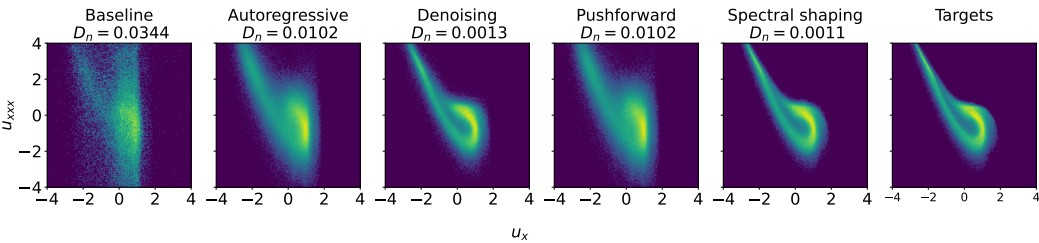

Figure 5: Phase space diagrams for the KS equation and various models. We plot 1[st] against 3[rd] derivatives. Above each figure we quote the 2-sample Kolmogorov-Smirnov distance $D_n$ from the observed histogram to the groundtruth target, essentially an $L_\infty$ norm on CDFs. Only denoising and spectral shaping have good correspondence with the targets. Full examples can be found in Appendix C.1, where spectral shaping clearly dominates.

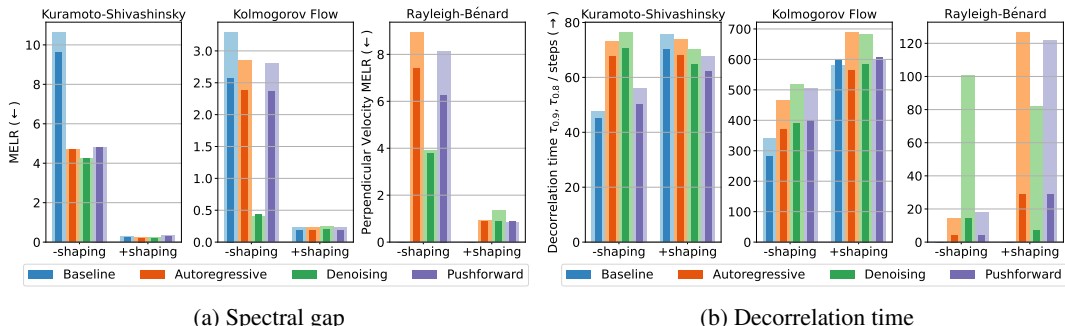

(a) Spectral gap
(b) Decorrelation time

Figure 6: Comparison with other stabilization techniques. Each bar represents the model that maximized decorrelation time $\tau_{0.8}$. Spectral gap is measured 128 (dark) & 256 (light) unroll steps after initial conditions. Reported decorrelation times are $\tau_{0.9}$ (dark) & $\tau_{0.8}$ (light). Filtering improves the spectrum and has a positive impact on the decorrelation window, lengthening it.

Appendix C.2, Figure 19. For RBC we also combined autoregressive training with all methods, since it is a difficult equation. For each bar, we selected the model that maximized decorrelation time. For results selected on minimum MELR see Appendix C.2.

As intended, we see that spectral shaping improves or maintains the spectrum alone (BASELINE) or combined with all techniques. Furthermore, although spectral shaping was only intended to improve the spectrum, we see in Figure 6(b) that it performs favorably against other techniques in terms of decorrelation time. The darker shaded bars represent $\tau_{0.9}$, and the lighter $\tau_{0.8}$. For the KS equation, we see that denoising, pushforward, and spectral shaping all perform similarly well, but on KF spectral shaping outperforms all competitors by a long way. This is a strong indicator that the improved spectrum is indeed useful for unroll accuracy. Critically spectral shaping allows us to expand the window of short-term correlation, and then after we have diverged from groundtruth, we continue to generate trajectories that have the correct spectral profile far off into the future.

**Vorticity PDFs** We can also inspect vorticity histograms, which are an indirect measure of spectral quality. Vorticity, the curl of velocity, is more sensitive to errors in the higher frequency band since derivatives weight energy content proportional to frequency. Note that these are 1D versions of the phase space plots in Section 5.1, where we view just a single derivative of the 2D velocity, the curl. We show results on RBC in Figure 7a, where the difference in histogram is visible to the naked eye. Spectral shaping improves over all baselines.

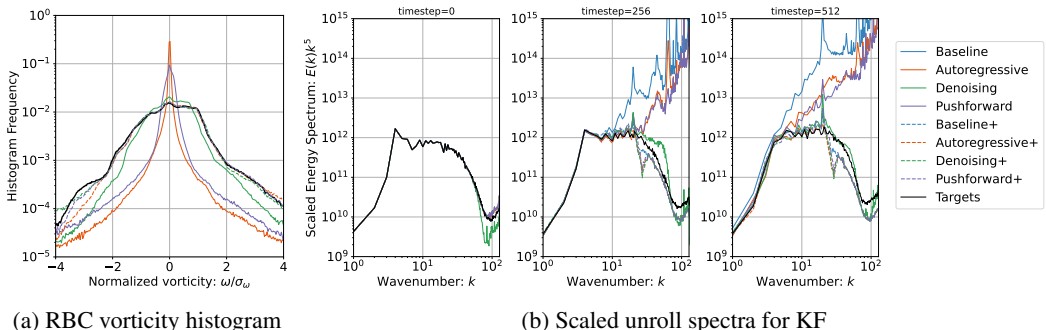

(a) RBC vorticity histogram
(b) Scaled unroll spectra for KF

Figure 7: (a) Histogram of normalized vorticities for Rayleigh-Bénard convection, averaged over 100 consecutive time steps. Spectral shaping (dashed lines) improves over its unfiltered counterparts (solid lines). $\sigma_\omega$ denotes the empirical standard deviation of histogrammed vorticities $\omega$. (b) Example rollout of spectrum on Kolmogorov flow. Although not perfect, we see that spectral shaping (dashed lines) vastly improves the spectrum in the mid- to high-frequency bands over unfiltered (solid lines) counterparts. That said, denoising provides a strong baseline.

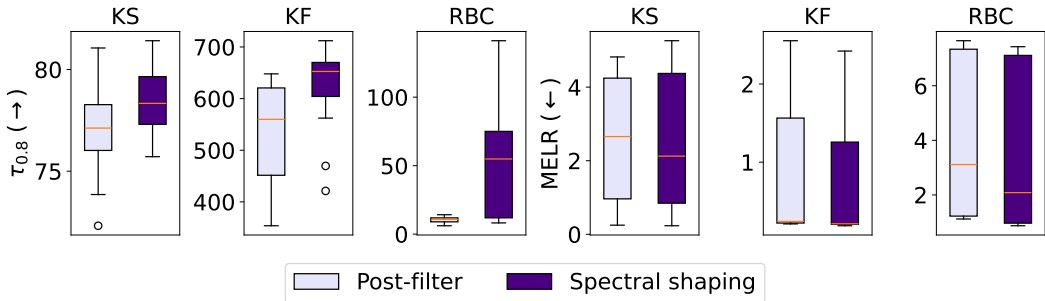

Figure 8: Comparison of spectral shaping (filtering after every nonlinearity) and post-filtering (low pass filtering the one-step model output only). We show box plots to highlight the range of attainable outcome metrics from a sweep over Gaussian widths. We see that while both methods can attain similar minimum spectral gaps, spectral shaping generally performs better on short-term accuracy, measured as decorrelation time $\tau_{0.8}$. This indicates the model requires filtering after every nonlinearity.

**Example spectra** Example spectra can seen in Figure 3a (KS), Figure 7 (KF), and Figure 20 (RBC). Figure 7 in particular shows the spectrum over an unroll. While the spectral-shaped spectrum diverges a little from the ground truth, the unshaped spectra diverge significantly. It should not be understated that the improvement in the high frequencies spans over 4 orders of magnitude for RBC, and 10 orders for KF and KS! The remain divergence for the spectral-shaped model could probably be remedied with a larger model or more training data.

**Could we just post-filter?** An obvious question is whether it is necessary to filter after *every* nonlinearity in our model? We run a simple test comparing filtering after every nonlinearity to filtering just at the output of the one-step models. We show box plots in Figure 8, to highlight the range of attainable outcome metrics from a sweep over Gaussian width. Notable is that post-filtering can attain minimal spectral gap, but generally poorer decorrelation time $\tau_{0.8}$, expecially on the tough problem of Rayleigh-Bénard convection. This indicates that spectral shaping deep inside a network aids its hidden representations for the task of unroll accuracy.

**Learning to minimize spectral gap** Another technique to improve spectral shape is to use a spectral gap minimizing loss. We explore two candidates: the MELR metric, and the DySLIM loss (Schiff et al., 2024). The DySLIM loss is a sum of two maximum mean discrepancy (Gretton et al., 2012) losses, one between a minibatch of predictions and a minibatch of targets, and another between the predictions and the initial conditions. We add each loss as a regularizer to the MSE loss, sweeping over hyperparameters to find a good combination. Results are shown in Figure 9. A combination of spectral shaping and a spectral loss always improves both spectrum and decorrelation window. This makes sense, because spectral shaping endows models the capacity to achieve sufficient dynamic range in the spectrum, while the spectral loss enforces the correct final shape.

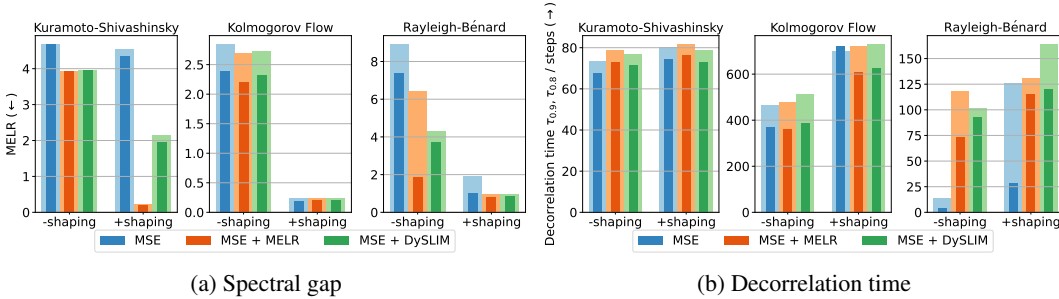

(a) Spectral gap          (b) Decorrelation time

Figure 9: Combinations of spectral shaping and spectral losses. All models are trained autoregressively, with up to 5 training time unrolls. In all cases a combination of shaping and spectral losses improves the spectrum and decorrelation time over shaping or optimization alone.

## 6 CONCLUSIONS

We have conducted an analysis into the role and importance of the spectrum on long-term temporal rollouts of deterministic autoregressive PDE surrogates. As corroborated by other works (Lippe et al., 2023), we found poor modeling of the mid- to high-frequency modes limits the extent of rollout accuracy. We identified three convergent factors accountable for this poor mid-to-high frequency modeling: statistical bias of training data versus test time rollouts (the distribution shift problem (Brandstetter et al., 2022)), learning dynamics bias encouraging low frequencies to be learned first (spectral bias of the L2 loss (Rahaman et al., 2019)), and architectural bias injecting high amounts of spectral junk (pointwise nonlinearities). We found that inserting a simple low pass filter after every pointwise nonlinearity in the model was enough to address the poor spectrum and improve rollout accuracy. Thus a simple modification to training can significantly improve PDE surrogate modeling and rollout capabilities.

We demonstrated this on three systems in 1D and 2D, exhibiting a range of behaviors, including chaos, non-stationarity, and non-periodic boundary conditions. We also showed that low pass filtering cannot easily be learned by the addition of a high frequency regularizer to the loss function: a (potentially learnable) low pass filtering operation has to be placed strategically after every pointwise nonlinearity.

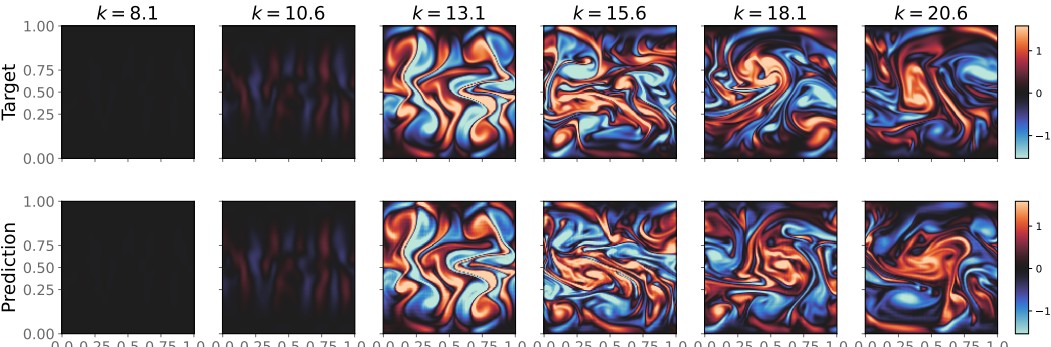

Figure 10: Example rollout from a spectrally shaped model on RBC. We plot the vorticity of the velocity fields only. We see that flow looks acceptable after decorrelating. This is because it has the correct rollout statistics in the form of energy spectrum.

## 7 ETHICS STATEMENT

This paper presents a simple technique to improve the long-term unroll quality of neural PDE surrogates. Surrogates as a whole have the potential to move us toward computationally cheaper simulators. Cheaper simulators implies cheaper design and increased industrial productivity, throughput, and workflows. As with all technologies, simulators are dual use. As to the ethical implications of this work, the authors consider this methodological treatise too far removed from downstream impacts for its ethical nature to be evaluated properly.

## 8 REPRODUCIBILITY STATEMENT

We report the model architecture used in Appendix B along with training settings. The exact method introduced can be found in equations 3 and 4. Details on how to generate each dataset can be found in Appendix A. Exact data and training performance may vary depending on random number seed chosen. The method is so simple, we encourage readers to try it out on their own problems!

Our implementation is written in JAX (Bradbury et al., 2018). We use MATPLOTLIB for plotting, NUMPY (Harris et al., 2020) and PANDAS (pandas development team, 2020) for data handling. All training runs were performed on an array of 8 NVIDIA V100 GPU, with trivial data parallelism achieved with `jax.pmap`. A single NVIDIA P100 GPU was used for evaluation. Training never took longer than one day.

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

## A DATASETS

### A.1 KURAMOTO-SHIVASHINSKY EQUATION

**Equation** The Kuramoto-Shivashinsky equation is a fourth-order chaotic 1D equation. Being chaotic, solutions are extremely sensitive to uncertainty in the initial conditions, and indeed any errors accumulated in the forward unroll. The PDE, in derivative form[2], is defined

$$u_t = -u_{xx} - u_{xxxx} - uu_x, \qquad x \in [0, 32\pi) \tag{6}$$

We impose periodic boundary conditions. Despite being 1D, this equation is very tough to work with due to the presence of the destabilizing negative diffusion term and the chaos-inducing hyper-diffusion term.

**Data generation** We generate trajectories using a modified version of the exponential time-differencing fourth-order Runge–Kutta method (ETDRK4) as found in Kassam & Trefethen (2005). Initial conditions are sampled from a mixture of sinusoids similarly to Bar-Sinai et al. (2019) of the form

$$u^0(x) = \sum_{k=1}^{K} A_k \sin(2\pi \ell_k x / L + \phi_k) \qquad x \in \Omega = [0, L]. \tag{7}$$

Domain $\Omega$ is discretized on to a uniform grid $\mathbb{X}$ with resolution of 512 points and periodic boundaries. The parameters are shown in Table 1. Note that while the solver uses a timestep of 1 ms, found from a convergence analysis, we only store every 100th step at 0.8 s intervals.

Table 1: Parameters for generating the KS dataset

| | |
|---|---|
| Number of modes | $K = 10$ |
| Amplitudes | $A_k \sim \text{Uniform}_{\mathbb{R}}(-0.5, 0.5)$ |
| Wavenumbers | $\ell_k \sim \text{Uniform}_{\mathbb{Z}}(1, 10)$ |
| Phases | $\phi_k \sim \text{Uniform}_{\mathbb{R}}(0, 2\pi)$ |
| Domain size | $L = 32\pi$ |
| Num grid points | $N = 512$ |
| Grid resolution | $\Delta x = L/N$ |
| Solver time step | $\Delta t_{\text{solver}} = 0.008s$ |
| Stored time step | $\Delta t = 0.8s$ |
| Burn in time | $t = 102.4$ s |
| Num training trajectories | 65536 |
| Training trajectory length | 64 steps |
| Num valid/test trajectories | 1024 |
| Valid/test trajectory length | 1024 steps |
| `float64` | `True` |

---

[2]Not to be confused with the integral form $v_t = -v_{xx} - v_{xxxx} - \frac{1}{2}v_x^2$, obtained by setting $v = u_x$.

## A.2 KOLMOGOROV FLOW

**Equation**  Kolmogorov flow (KF) refers to 2D forced incompressible Navier-Stokes. The PDE is defined on a toroidal domain $\Omega = [0, 2\pi)^2$ as

$$\mathbf{u}_t = -\nabla \cdot (\mathbf{u} \otimes \mathbf{u}) + \frac{1}{\mathrm{Re}} \nabla^2 u - \frac{1}{\rho} \nabla p + \mathbf{f} \tag{8}$$

$$\nabla \cdot \mathbf{u} = 0 \tag{9}$$

for velocity field $\mathbf{u}(x, y) \in \mathbb{R}^2$ defined at coordinates $(x, y) \in \Omega$, tensor product $\otimes$, constant density $\rho$, pressure $p$, Reynolds number $\mathrm{Re} \gg 1$, and peak wavenumber $k_0 = 4$ and forcing function

$$\mathbf{f} = \begin{bmatrix} 0 \\ \sin(k_0 y) \end{bmatrix} + 0.1. \tag{10}$$

In our experiments we set $\mathrm{Re} = 500$, which exhibits strong roll off in the spectral domain.

**Data generation**  We generate data using the JAX-CFD finite volume code from Kochkov et al. (2021a). Details concerning the solver settings can be found in Table 2. The data is discretized on to a uniform 2D grid with periodic boundaries.

Table 2: Parameters for generating the Kolomogorov flow dataset

| | |
|---|---|
| Reynolds number ($Re$) | 500 |
| Maximum velocity | 7 |
| Peak wavenumber | 4 |
| Maximum Courant number | 0.5 |
| Domain size | $(L_x, L_y) = (2\pi, 2\pi)$ |
| Num grid points (generation) | $(N_x, N_y) = (512, 512)$ |
| Num grid points (storage) | $(N_x, N_y) = (256, 256)$ |
| Grid resolution | $(\Delta x, \Delta y) = (2\pi/256, 2\pi/256)$ |
| Solver time step | $\Delta t_{\text{solver}} = 0.0008s$ |
| Stored time step | $\Delta t = 0.1s$ |
| Burn in time | $t = 10$ s |
| Num training trajectories | 8192 |
| Training trajectory length | 16 steps |
| Num valid/test trajectories | 128 |
| Valid/test trajectory length | 1024 steps |
| `float64` | `True` |

### A.3 RAYLEIGH-BÉNARD CONVECTION

**Equation** *Rayleigh-Bénard convection* (RBC) is a model of turbulent thermal convection, where a fluid is confined between two plates, heated below and cooled from above. This phenomenon is tricky to model, because it contains non-periodic boundary conditions at the two plates, turbulent decay, non-stationarity, and extreme sensitivity to initial conditions. The equations are

$$\mathbf{u}_t = -(\mathbf{u} \cdot \nabla)\mathbf{u} + \sqrt{\frac{P}{R}}\nabla^2\mathbf{u} - \nabla p + \theta\mathbf{e}_z \tag{11}$$

$$\theta_t = -(\mathbf{u} \cdot \nabla)\theta + \frac{1}{\sqrt{PR}}\nabla^2\theta \tag{12}$$

$$\nabla \cdot \mathbf{u} = 0 \tag{13}$$

for domain $[x, z] \in [0, 1]^2$. The boundary conditions at the upper and lower plates are

$$\theta(z = 0) = 1, \theta(z = 1) = 0, \mathbf{u}(z = 0) = \mathbf{0}, \mathbf{u}(z = 1) = \mathbf{0}. \tag{14}$$

We have defined temperature $\theta$ (also referred to as buoyancy), vertical unit vector $\mathbf{e}_z$, *Prandtl number* $P$ (= 1 in all our experiments), and *Rayleigh number* $R \gg 1$. The boundary conditions represent the constant temperature plates and *no-slip* condition on the fluid velocity. To make this even harder, the equations are parametrized by Rayleigh number Ra, which we sample from a log-uniform distribution and pass as side-information to our models. The Rayleigh number is a dimensionless number representing the balance between buoyancy-driven flow and viscous/thermal diffusion. The higher it is the more mixing we expect to see.

**Data generation** We generate data using the Dedalus spectral solver from Burns et al. (2020). We use a modified version of the Rayleigh-Bénard convection code found at `https://github.com/DedalusProject/dedalus/blob/master/examples/ivp_2d_rayleigh_benard/rayleigh_benard.py`. Details concerning the solver settings can be found in Table 3. We pick the range of Rayleigh numbers (Ra) in the turbulent unstable range, as high as we can go before the solver can no longer satisfy the incompressibility condition $\nabla \dot{\mathbf{u}} = 0$. The data is discretized on to a 2D grid with uniform sampling in the horizontal (periodic) direction and Chebyshev root point sampling in the vertical direction with boundary; that is

$$(x_i, z_j) = (i\Delta x, 0.5 \cdot (\cos(\pi(j + 0.5)/N) + 1)), \qquad i, j \in 0, 1, 2, ..., N - 1. \tag{15}$$

For initial conditions we use zero velocity field and noisy linear temperature ramp with normalized temperature 1 unit on the bottom plate and 0 units on the top plate.

$$\mathbf{u}^0(x_i, z_j) = \mathbf{0} \tag{16}$$

$$\theta^0(x_i, z_j) = (1 + \sigma\epsilon_j z_j)(L_z - z_j). \tag{17}$$

We burn in for 8 s and then keep the remaining 25 s for train/valid/test splits. For validation/test trajectories we use the full 25 s, discretized into 400 steps. For training, we split the 400 steps into 25 trajectories of 16 steps each.

Table 3: Parameters for generating the Kolmogorov flow dataset

| | |
|---|---|
| Rayleigh number | Ra $\in$ Log-uniform$(10^6, 10^7)$ |
| Prandtl number | Pr = 1 |
| Initial condition noise | $\sigma = 10^{-3}$ |
| Domain size | $(L_x, L_z) = (1, 1)$ |
| Num grid points (generation/storage) | $(N_x, N_z) = (256, 256)$ |
| Grid | $(x_i, z_j) = (i/256, 0.5 \cdot (\cos(\pi(j + 0.5)/256) + 1))$ |
| Solver time step | Adaptive |
| Stored time step | $\Delta t = 0.0625s$ |
| Burn in time | $t = 8$ s |
| Num training trajectories | 12800 |
| Training trajectory length | 16 steps |
| Num valid/test trajectories | 256 |
| Valid/test trajectory length | 400 steps |
| `float64` | `True` |

## B  MODEL SPECIFICATION

For simplicity, we use the same training settings across all experiments.

Table 4: UNet architecture. conv($3^d$, 64) denotes a convolution with kernel size 3 in $d$ spatial dimensions, and 64 output channels. We use a modified version of the basic UNet (Ronneberger et al., 2015) with nearest neighbor upsampling instead of transpose convolutions as per Odena et al. (2016). If spectral shaping is applied, it is placed after every softplus nonlinearity. The softplus nonlinearity is chosen for having the smoothest spectrum among nonlinearities offered in JAX.

| Layer number | Type |
|:---:|:---:|
| | Encoder |
| 1 | [pad $\rightarrow$ conv($3^d$, 32) $\rightarrow$ softplus $\rightarrow$ layer norm] |
| 2 | [pad $\rightarrow$ conv($3^d$, 32) $\rightarrow$ softplus $\rightarrow$ layer norm] |
| 3 | average pool(kernel size $2^d$, stride $2^d$) |
| 4 | [pad $\rightarrow$ conv($3^d$, 64) $\rightarrow$ softplus $\rightarrow$ layer norm] |
| 5 | [pad $\rightarrow$ conv($3^d$, 64) $\rightarrow$ softplus $\rightarrow$ layer norm] |
| 6 | average pool(kernel size $2^d$, stride $2^d$) |
| 7 | [pad $\rightarrow$ conv($3^d$, 128) $\rightarrow$ softplus $\rightarrow$ layer norm] |
| 8 | [pad $\rightarrow$ conv($3^d$, 128) $\rightarrow$ softplus $\rightarrow$ layer norm] |
| 9 | average pool(kernel size $2^d$, stride $2^d$) |
| | Bottleneck |
| 10 | [pad $\rightarrow$ conv($3^d$, 256) $\rightarrow$ softplus $\rightarrow$ layer norm] |
| 11 | [pad $\rightarrow$ conv($3^d$, 256) $\rightarrow$ softplus $\rightarrow$ layer norm] |
| | Decoder |
| 12 | nearest neighbor upsample($2^d$) |
| 13 | concatenate(output 12, output 8) |
| 14 | [pad $\rightarrow$ conv($3^d$, 32) $\rightarrow$ softplus $\rightarrow$ layer norm] |
| 15 | [pad $\rightarrow$ conv($3^d$, 32) $\rightarrow$ softplus $\rightarrow$ layer norm] |
| 16 | nearest neighbor upsample($2^d$) |
| 17 | concatenate(output 16, output 5) |
| 18 | [pad $\rightarrow$ conv($3^d$, 64) $\rightarrow$ softplus $\rightarrow$ layer norm] |
| 19 | [pad $\rightarrow$ conv($3^d$, 64) $\rightarrow$ softplus $\rightarrow$ layer norm] |
| 20 | nearest neighbor upsample($2^d$) |
| 21 | concatenate(output 20, output 2) |
| 22 | [pad $\rightarrow$ conv($3^d$, 128) $\rightarrow$ softplus $\rightarrow$ layer norm] |
| 23 | [pad $\rightarrow$ conv($3^d$, 128) $\rightarrow$ softplus $\rightarrow$ layer norm] |
| 24 | conv($1^d$, num fields) |

Table 5: Training settings. Values of the form $x_{1D}/x_{2D}$ use $x_{1D}$ for the 1D experiments and $x_{2D}$ for the 2D experiments, otherwise the same settings are used for both 1D and 2D.

| Setting | Values |
|:---:|:---:|
| Batch size | 1024 / 64 |
| Roll-in length per AR step | 1 |
| Roll-out length per AR step | 1 |
| Precision | `float64/float32` |
| Optimizer | ADAMW($b_1 = 0.9, b_2 = 0.999$, weight decay $= 1e-5$) |
| Learning rate schedule (part I) | Constant(`1e-3`, 200k steps) |
| Learning rate schedule (part II) | CosineDecay(`1e-3` $\rightarrow$ `1e-4`, 200k steps) |

## C  Supplementary visualizations

### C.1  Phase space diagrams

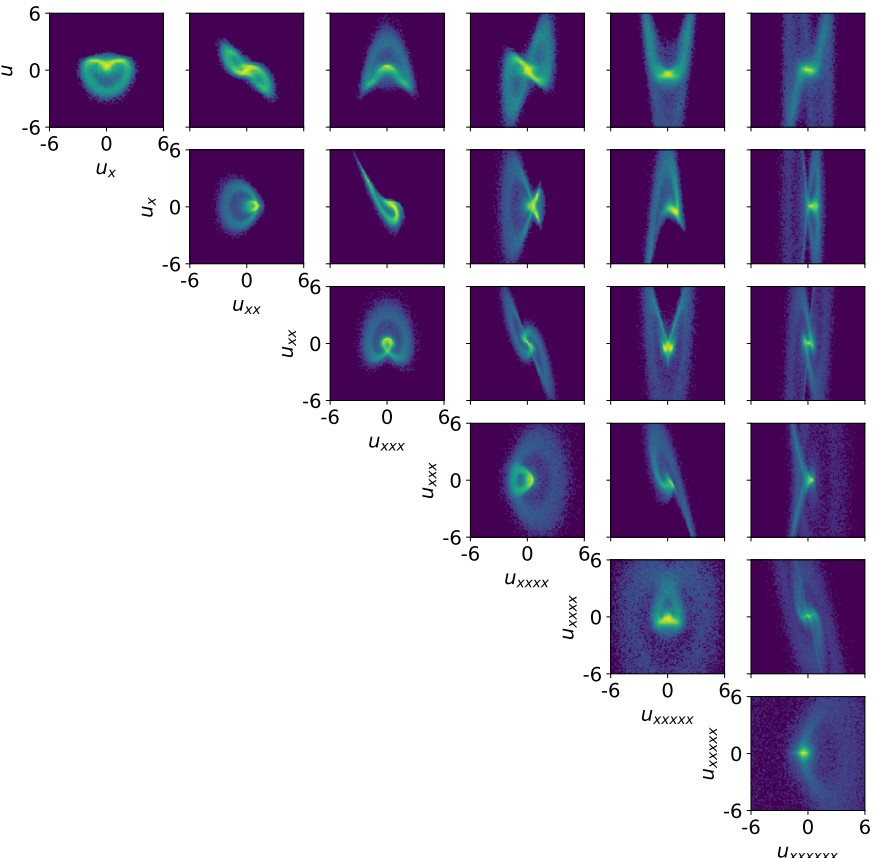

Figure 11: Phase space plots of the targets of the KS equation. Shown for reference.

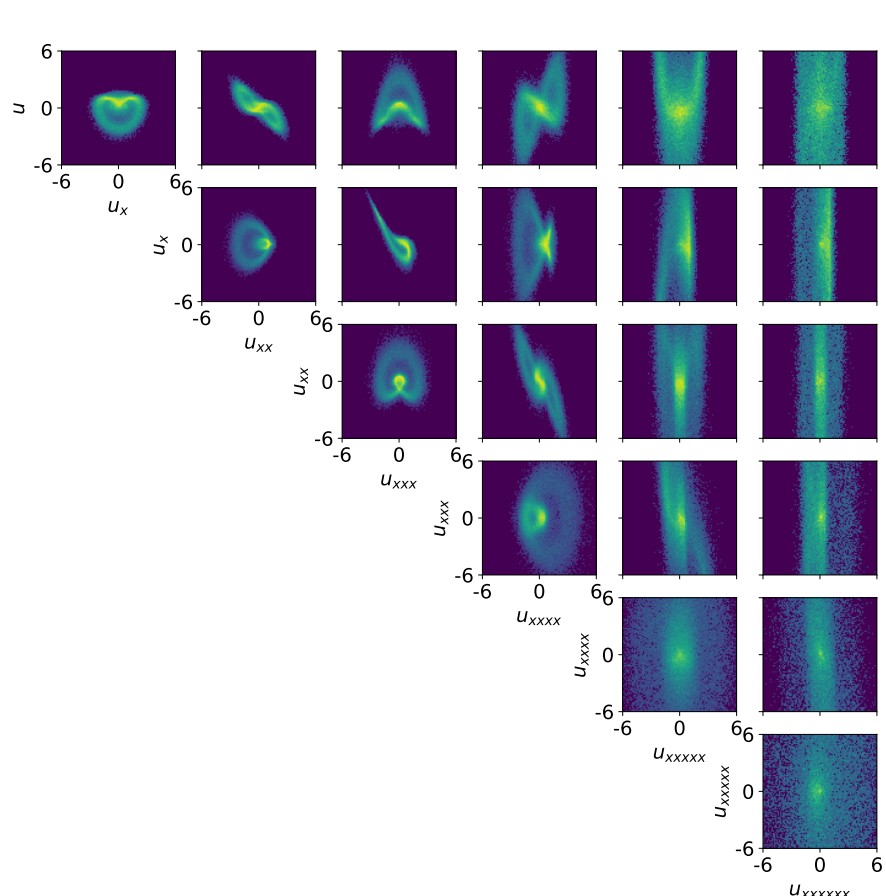

Figure 12: Phase space plots for spectral shaping on the KS equation. Notice how it is the only method to capture higher order derivative statistics.

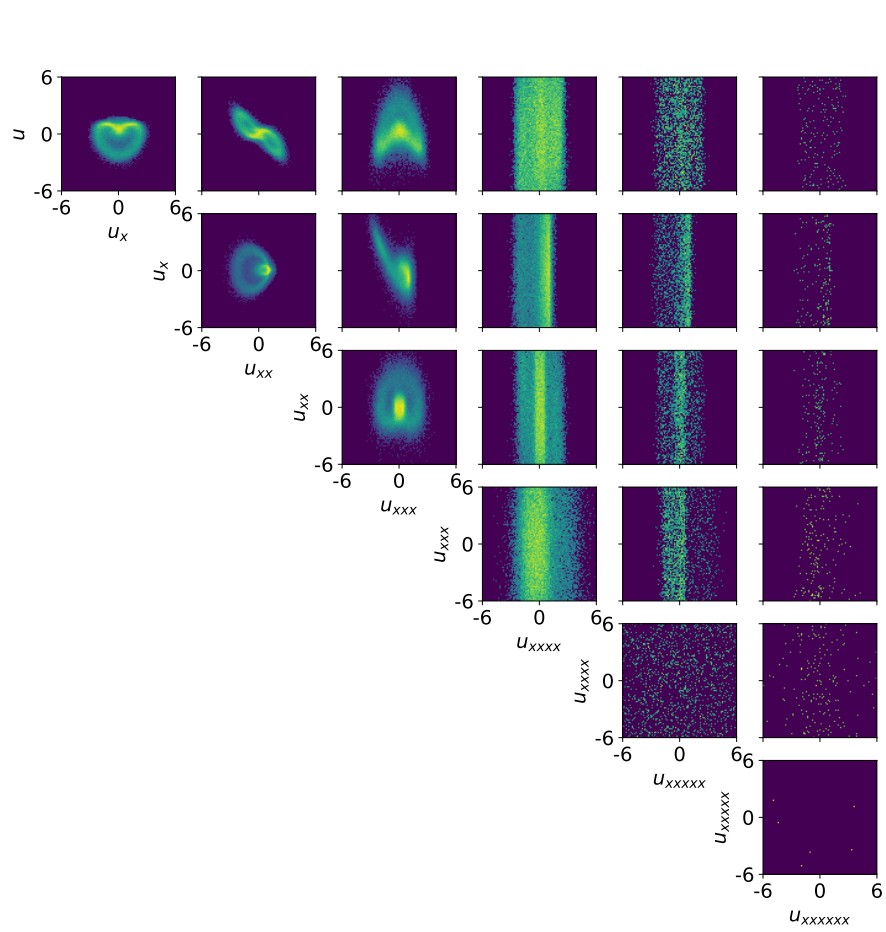

Figure 13: Phase space plots for the pushforward trick on the KS equation.

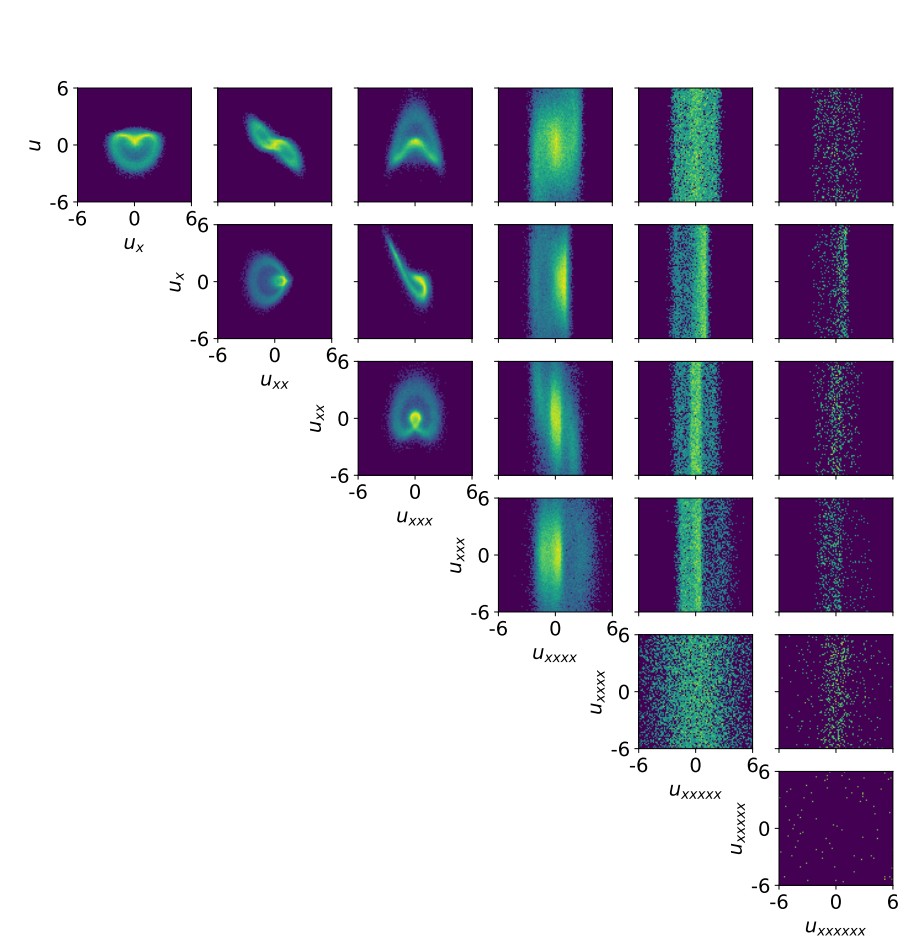

Figure 14: Phase space plots for denoising on the KS equation.

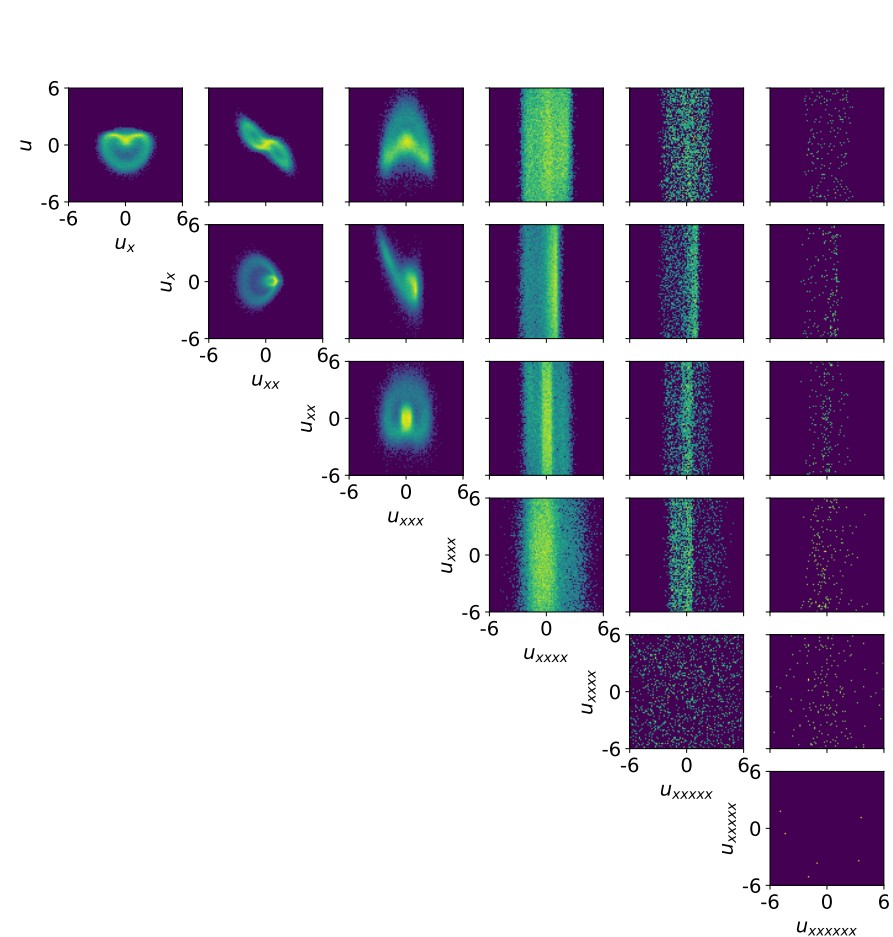

Figure 15: Phase space plots for autoregressive training on the KS equation.

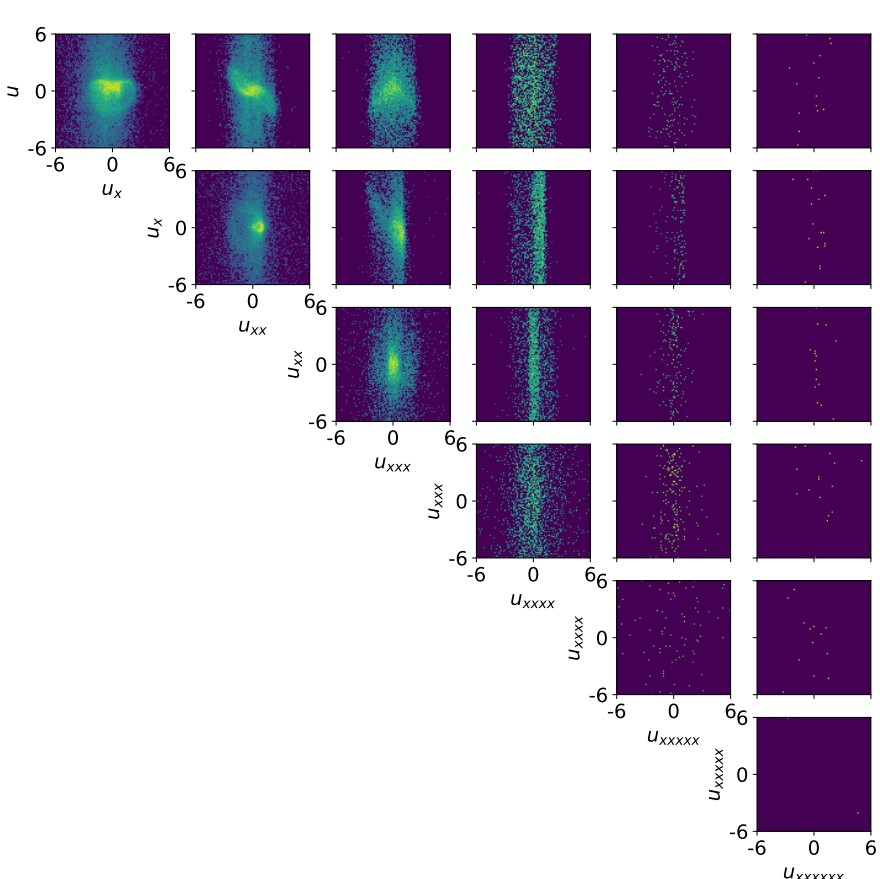

Figure 16: Phase space plots for baseline on the KS equation.

## C.2 EXTRA COMPARISONS

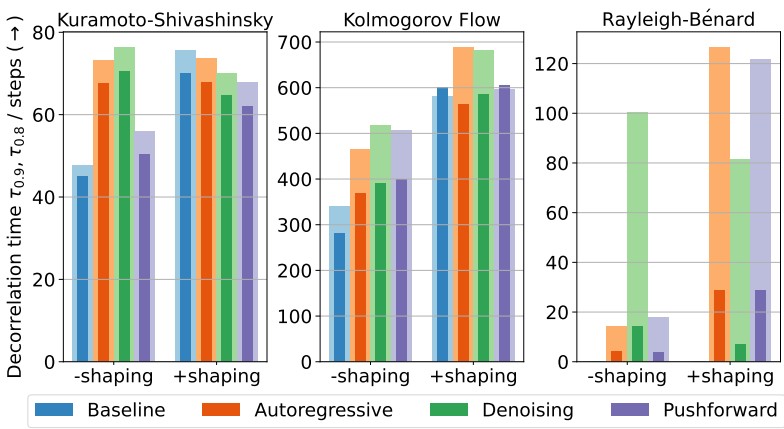

Figure 17: Decorrelation time selecting on minimum MELR (perpendicular velocity MELR for RBC).

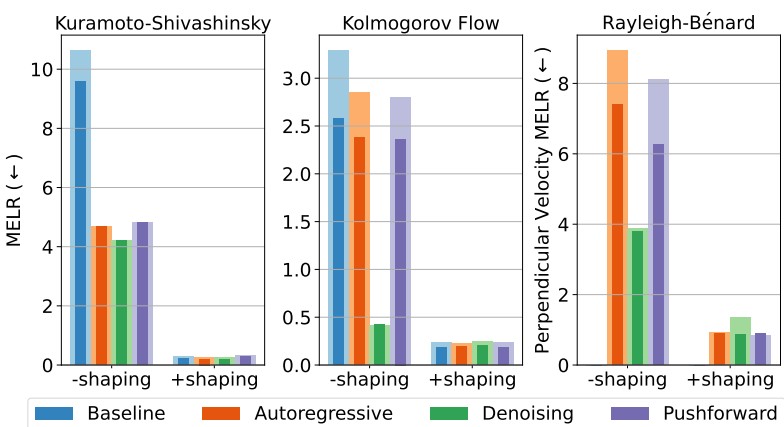

Figure 18: MELR selecting on minimum MELR (perpendicular velocity MELR for RBC.

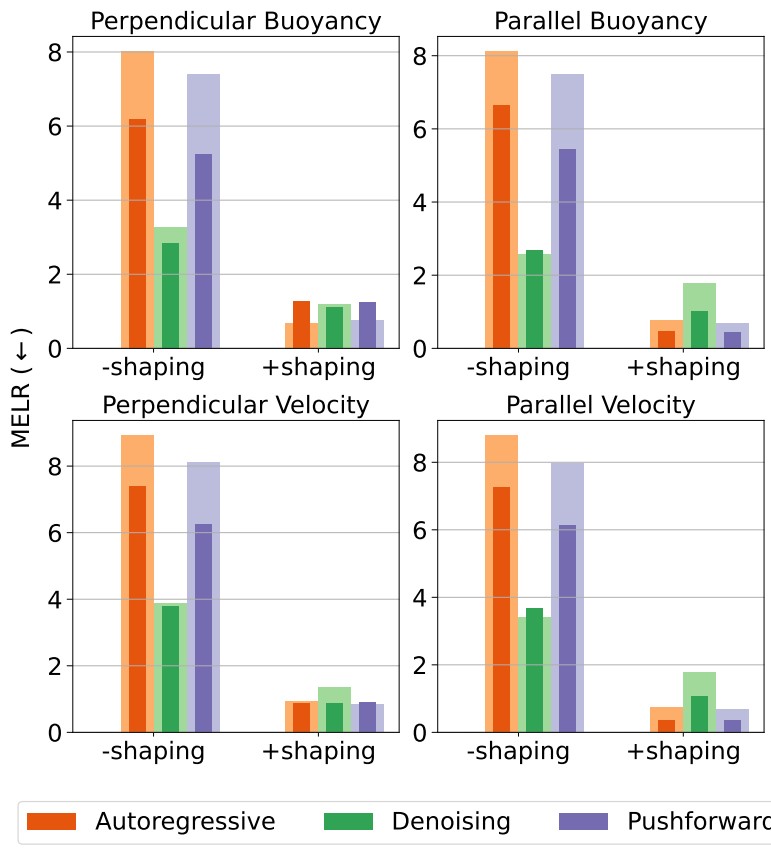

Figure 19: MELR for all spectra on RBC, selecting on maximum decorrelation time. We see the typical pattern that spectral shaping improves MELR for all methods. Furthermore, the improvement in spectrum is independent of output field (buoyancy/velocity) and how we average it (parallel/perpendicular).

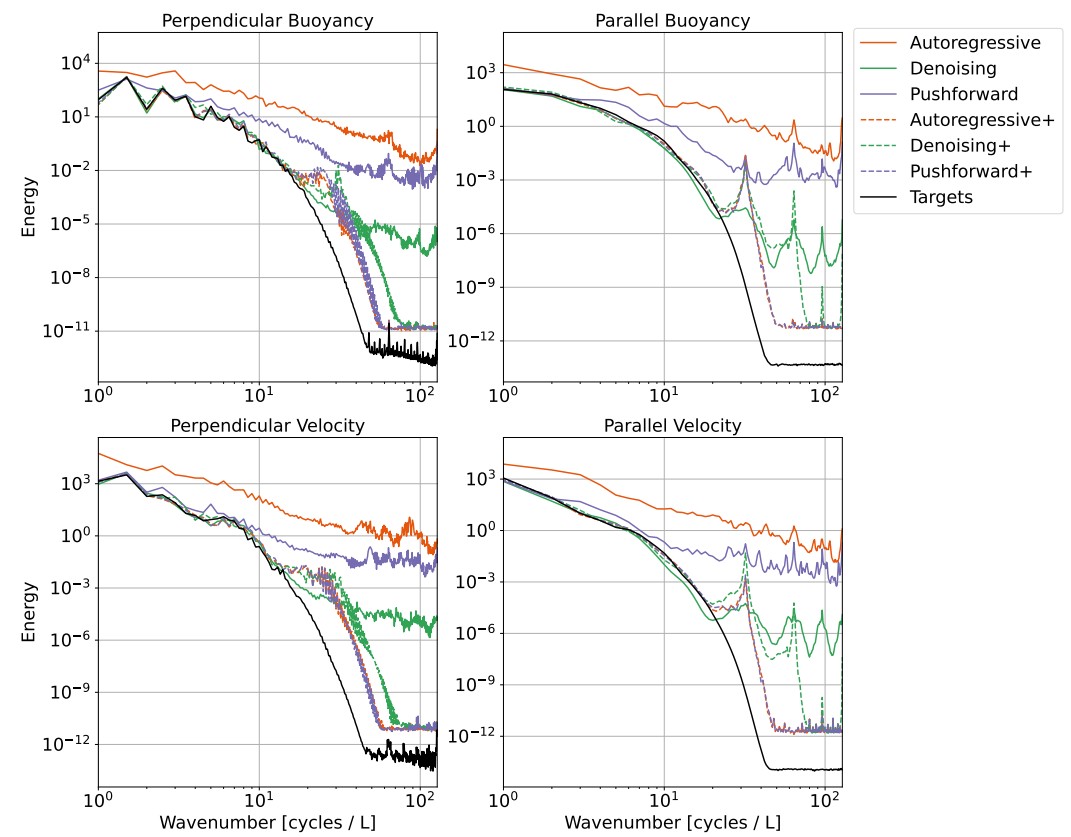

Figure 20: Energy spectra for the best performing models on the Rayleigh-Bénard convection problem. The models correspond to those reported in Figure 6a. Although not perfect, we see that spectral filtering (dashed lines) significantly improves over the unfiltered counterparts (solid lines), between 6 - 10 orders of magnitude in the high frequency band.

## D  COMPUTING SPECTRA

The 1D energy spectrum for Kuramoto-Shivashinsky is computed as

$$E_{\text{KS}}(k) = \left| \sum_{n=1}^{N} u_n \cdot e^{-i2\pi \frac{k}{N} n} \right|^2.$$ (18)

We only consider the 1-sided spectrum $k \geq 0$, due to conjugate symmetry.

For Kolmogorov Flow and Rayleigh-Bénard convection we compute direction-averaged energy spectra. Given 2D energy spectrum $E(\mathbf{k})$, for $(k_1, k_2) = \mathbf{k} \in \mathbb{R}^2$, the direction averaged energy spectrum is

$$E_{\text{average}}(k) = \frac{1}{|S_k|} \sum_{\mathbf{k} \in S_k} E_{\text{KF}}(\mathbf{k})$$ (19)

where $S_k$ is the set of equivalent frequencies. We consider the three scenarios:

- Radial averaging: $S_k = \{\mathbf{k} \in \mathbb{R}^2 \mid |\mathbf{k}| = k\}$ (average over $\mathbf{k}$ of equal radius)

- Parallel averaging: $S_k = \{\mathbf{k} \in \mathbb{R}^2 \mid k_1 = k\}$ (average over $k_2$, direction of equal parallel direction with respect to plates)

- Perpendicular averaging: $S_k = \{\mathbf{k} \in \mathbb{R}^2 \mid k_2 = k\}$ (average over $k_1$, direction of equal perpendicular distance to plates)

The 2D energy spectrum of Kolmogorov flow is computed as

$$E_{\text{KF}}(k_1, k_2) = \left| \sum_{n_1=1}^{N} \sum_{n_2=1}^{N} u_{n_1, n_2} \cdot e^{-i2\pi \frac{k_1}{N} n_1} e^{-i2\pi \frac{k_2}{N} n_2} \right|^2, \tag{20}$$

which is the 2D DFT over the domain. The 2D energy spectrum for Rayleigh-Bénard convection is computed as

$$E_{\text{RBC}}(k_1, k_2) = \left| \sum_{n_1=1}^{N} \sum_{n_2=0}^{N-1} u_{n_1, n_2} \cdot e^{-i2\pi \frac{k_1}{N} n_1} \cos\left( \frac{\pi}{N} \left( n_2 + \frac{1}{2} \right) k_2 \right) \right|^2, \tag{21}$$

which is a DFT over $n_1$, the periodic direction, and a type-II DCT (Makhoul, 1980) over $n_2$, the non-periodic direction.

