# OpenReview forum: "Spectral Shaping for Neural PDE Surrogates"
_ICLR.cc/2025/Conference — Submitted to ICLR 2025_

### Official Review · Reviewer_TbPC · 2024-10-30

**Soundness:** 2
**Presentation:** 3
**Contribution:** 2
**Rating:** 5
**Confidence:** 4

**Summary:**

This paper addresses problems with using neural PDE solvers, specifically the spectrum of the solution in the mid-high frequency band.  The authors present the key findings of the paper as the discovery that  higher frequencies are introduced by point-wise nonlinearities and address this issue experimentally with the application of a Gaussian low pass filter after every non-linearity.

**Strengths:**

Overall, the paper does a good job of presenting the issue within the context of neural PDE solvers and the literature around using machine learning to solve PDEs is reasonably well surveyed.

I am pleased to see the area of spectral shaping applied to neural PDE solvers being explored, and I think that the results in the paper show that it is a promising area.

If a thorough evaluation of the literature on both sides of the work (i.e. both spectral shaping and neural PDE solvers) is included then this will enrich the insights in the paper and has the potential to be a worthy addition to the literature in this area.

**Weaknesses:**

**Main Comments**

In contrast with the literature on neural PDE solvers, the background literature around the effect of nonlinearities on the spectrum and filtering has not been sufficiently explored. This is a major drawback of the paper and something that should be improved in order to accept it for publication and a number of related papers are referenced at the end of this review. In order to make this work worthy of publication, a much more thorough evaluation of the literature around spectral shaping should be carried out and therefore, in its current form, I would lean towards rejecting the paper unless the following points can be addressed.

This is especially true because a survey of some of the literature around this area will not only provide a more complete background to the work, but also give a theoretical underpinning to some of the phenomena that are described.
For example from the “integration by parts coefficient bound” described by [1], if the function is smooth up to and including its $(k-2)^\text{th}$ derivative, and its $k^\text{th}$ derivative is integrable, then the Fourier coefficients decrease as $O(\xi^{-k})$. This provides a possible reason for the increase in spectrum at mid and high frequencies due to discontinuities. In particular, I am skeptical that the post-nonlinearity increase observed at higher frequencies is indeed "noise" as the authors describe, instead they may be due to the change in spectral shape due to discontinuities. In addition, the excellent survey by [5] also links these effects back to the original observation by Gibbs [2,3] regarding the interaction between discontinuities in the time (frequency) domain and the shape in the frequency (time) domain.  In order to provide evidence for this the authors could experimentally examine the relationship between the order of the discontinuity and the shape of the spectrum.

The work by [1] also provides a motivation for the requirement for the filters to be applied after each stage, rather than once at the end; as the signal energy at the higher frequencies increases due to discontinuities, the effect of truncating the higher frequencies will be greater. This effect can be assessed by calculating the error that will be introduced by the discontinuities and comparing that to experimental observations. The work in [1,4] shows some example calculations of errors.

In addition to this background literature on the effect of discontinuities, some of the more significant literature on the subject of filtering should be included, e.g. [4,6,7]. This can provide some theoretical justification to the ultimate selection of the Gaussian filter, such as filter orders and experimental results from testing.

[1] J. P. Boyd. Chebyshev and Fourier Spectral Methods. Springer, Heidelberg, 2001.

[2] J. W. Gibbs. Fourier’s series. Nature, 59:200, 1898.

[3] J. W. Gibbs. Fourier’s series. Nature, 59:606, 1899.

[4] D. Gottlieb and C.-W. Shu. On the Gibbs phenomenon and its resolution. SIAM Review, 39(4):644–668, 1997.

[5] E. Hewitt and R. E. Hewitt. The Gibbs-Wilbraham phenomenon: an episode in Fourier analysis. Archive for History of Exact Sciences, 21(2):129–160,1979.

[6] E. Tadmor. Filters, mollifiers and the computation of the Gibbs phenomenon. Acta Numerica, 16:305–378, 2007.

[7] H. Vandeven. Family of spectral filters for discontinuous problems. Journal of Scientific Computing, 6(2):159–192, 1991.

**Minor Comments**

In general, the paper is well formatted and written. However, I have a few minor comments:

In acronyms are used before they are defined, e.g. the use of KS in the caption of Figure 3 before the use of Kuramoto-Shivashinsky in the text.

In some cases the language could be slightly more formal in an academic paper. For example the use of ``sanity check"  and "probably" in Section 5.1.

Also in Section 5.1, I would like a more formal justification for the statement "The remain [sic] divergence for the spectral-shaped model could probably be remedied with a larger model or more training data." I think it likely that the addition theoretical work suggested in Section 2 of this review will also inform this improvement.

**Questions:**

I would be pleased to hear the authors' feedback on my suggestions above. I think this could be a useful area of work once more work has been done on the theory around discontinuities and spectral shaping.

---

> ### Author Response · Authors · 2024-11-25
> **Regarding the origin of spectral junk**
>
> Thanks for your comments and requests for a more in-depth theoretical treatment of how nonlinearities introduce spectral junk. We can certainly strengthen our exposition with a detailed mathematical description of the mechanism whereby it is introduced. We can explain the phenomenon through amplitude distortion. Mathematically what happens is as follows:
>
> Consider a scalar-valued function $u(x)$ and a pointwise nonlinearity $f: \mathbb{R} \to \mathbb{R}$, such that $v(x) = f(u(x))$. We will study how the Fourier transform $V(k)$ of $v(x)$ for wavenumber $k$ is related to the Fourier transform $U(k)$ of $u(x)$.
>
> To do this we approximate $f$ with its Taylor expansion about zero (we assume f is smooth) as
>
> $$v(x) = f(u(x)) = f(0) + u(x) \cdot f’(0) + u(x)^2 \cdot f’’(0) / 2! + u(x)^3 \cdot f’’’(0) / 3! + \text{h.o.t.}$$
>
> Then note how the nonlinear terms are monomials. The Fourier transform of $u(x)^2$ is the autoconvolution $U * U$, where $*$ denotes convolution. For the nth power, we have the convolution $U$ with $n-1$ copies of itself. This serves to broaden out the spectrum, mixing high and low frequencies of the spectrum and makes the spectrum more Gaussian in shape (indeed this is the same reason why sums of random variables undergo Gaussianization in central limit theorems, since sums of random variables are convolutions of their PDFs). For smooth signals $u(x)$ most of the energy is in the low frequencies. Passing $u(x)$ through a nonlinearity $f$ moves some of the energy through to the higher frequencies through this autoconvolutional mechanism. This is why we see an increase in power in the higher frequencies of the spectrum.
>
> Note that we do not require there to be any discontinuities for this effect to be seen and in the above treatment all the operations were smooth. Indeed in our submission all PDEs are smooth and all architectures are composed of smooth operations. Regarding our comment about spectral junk being “noise”, we can tighten up our phrasing here and reword this as interference of the signal with itself, which is exactly what this is. Indeed this phenomenon is different to aliasing, being induced by a different mechanism.
>
> We hope this clears up any confusion concerning the origin of spectral junk. We shall include this in an updated draft of the paper.
>
> **Minor points**
>
> * In acronyms are used before they are defined, e.g. the use of KS in the caption of Figure 3 before the use of Kuramoto-Shivashinsky in the text. **Thanks for spotting this! This is being corrected.**
>
> * In some cases the language could be slightly more formal in an academic paper. For example the use of ``sanity check" and "probably" in Section 5.1. **This is certainly personal preference, but we shall strive to update this.**

---

### Official Review · Reviewer_1CqM · 2024-11-01

**Soundness:** 2
**Presentation:** 3
**Contribution:** 2
**Rating:** 5
**Confidence:** 4

**Summary:**

The paper aims to improve the ability of neural PDE solvers to model high-frequency information accurately, which it claims is the reason for the long-term rollout instability of autoregressive models. The inability to model high-frequency behavior is attributed to the spectral bias of the MSE loss, distribution shift, and noise produced by non-linearities in the model. The authors conducted a study on the frequency behavior of the model, including an analysis of the eigen-spectrum. They propose to overcome the noise produced by the model by applying a fixed, preset noise filter after each non-linearity in the model. The experiments show that this helps the model to generate more accurate roll-outs which diverge from the true solution at a later point in time.

**Strengths:**

1. Novel analysis of the eigen-spectrum; including the insight that the unstable eigenvalues correspond to the high-frequency, noisy vectors
2. The proposed method improves the decorrelation times over the vanilla U-Net and can fix the high-frequency noise problem.
3. Easy-to-implement method

**Weaknesses:**

1. The presented study combines multiple known aspects of the rollout problem but has limited novelty on its own. Besides the eigen-spectrum analysis, most of the insights from the presented study have already been presented in other works. The inability to model high-frequency noise is a known problem and has been described in previous publications. (Lippe 2024, McCabe 2023). In McCabe et al. (2023), the noise problem was explicitly attributed to the non-linearity, and they proposed to learn to filter this noise out (although for FNO). In Raonić et al. (2024), formal solutions are introduced to avoid aliasing produced by non-linearities in convolutional models.
2. The proposed method is not compared against any other architectural methods that tackle the noise problem, e.g., PDERefiner, in the experiments. While the presented method is computationally cheaper, the question remains whether it can increase the decorrelation time as much as the other methods tackling the high-frequency noise problem.
3. The method is only evaluated for a single model (U-Net). This leads to the question of whether the good experimental results are due to using a model that may be especially prone to producing high-frequency noise. For example, the paper could be improved by evaluating whether the filter helps FNO. This would also allow a comparison to the method of McCabe et al. (2023).
4. The right filter curve is found by hand and not learned.

Lippe et al. "Pde-refiner: Achieving accurate long rollouts with neural pde solvers." Advances in Neural Information Processing Systems 36 (2024).
McCabe et al. "Towards Stability of Autoregressive Neural Operators." Transactions on machine learning research (2023).
Raonic et al. "Convolutional neural operators for robust and accurate learning of PDEs." Advances in Neural Information Processing Systems 36 (2024).

**Questions:**

1. Why did you choose to train the model in float64 precision?
2. What do the circles in the box-plot in Fig. 8 represent?

---

> ### Author Response · Authors · 2024-11-25
> **Rebuttal (1/2)**
>
> Thanks for the review. Please see our rebuttal and let us know if anything is missing. We have tried to address all your concerns. If you believe we have improved the exposition, please do upgrade your rating.
>
> **The presented study combines multiple known aspects of the rollout problem but has limited novelty on its own. Besides the eigen-spectrum analysis, most of the insights from the presented study have already been presented in other works. The inability to model high-frequency noise is a known problem and has been described in previous publications. (Lippe 2024, McCabe 2023). In McCabe et al. (2023), the noise problem was explicitly attributed to the non-linearity, and they proposed to learn to filter this noise out (although for FNO). In Raonić et al. (2024), formal solutions are introduced to avoid aliasing produced by non-linearities in convolutional models.**
>
> Thanks for pointing out those connections. The papers by Lippe, McCabe, and Raonić are all great works.
>
> Lippe et al. concerns itself with spectral shape, and we compare against their paper PDE-Refiner below. We find while they achieve good decorrelation time, so can we, with only a single pass through the model. We also achieve good spectral shape (low MELR), which implies better statistics in the far-term after we have rolled out far into the future.
>
> Raonić et al. concerns itself with aliasing. We do not. We are concerned with the introduction of spectral junk due to amplitude distortion from nonlinearities. Indeed the two processes are related and this is a well-known phenomenon, which we explain in detail to reviewer TbPC, which you may read at your leisure. The technique in the CNO paper by Raonić et al. is to oversample, but this only works for sufficiently bandlimited signals, which cannot be guaranteed with general nonlinearities. We accept this and opt for low pass filtering, instead.
>
> **The proposed method is not compared against any other architectural methods that tackle the noise problem, e.g., PDERefiner, in the experiments. While the presented method is computationally cheaper, the question remains whether it can increase the decorrelation time as much as the other methods tackling the high-frequency noise problem.**
>
> We have added some more baselines. But first an explanation why we did not initially include them. Models such as FNO, Galerkin Transformer, and other neural operator models are learned maps directly from initial conditions to solution values at time t, namely, $u(., 0) -> u(., t)$. Meanwhile the focus of this paper is on autoregressive methods. That said, nothing stops us from training a Neural Operator model in an autoregressive fashion and we do so for the rebuttals, but this is really not the arena where such models shine. Regarding Galerkin Transformer, notions of spatial geometry are lost in the attention layer and so it is not obvious how to apply spectral shaping in that space. PDE-Refiner, a diffusion-type model, is stochastic and was not the focus of this paper, but we include it, since it is state-of-the-art. It cannot be combined with spectral shaping.
>
> We provide these for the KS equation, which is all we had time for in the rebuttal period. We report MELR (Mean energy log ratio, a measure of spectral shape) at 128 time steps (lower is better) and decorrelation time  at 0.8 Pearson’s correlation coefficient (higher is better).
>
> (**Bold** indicates best in each row)
>
> | Model | MELR (no shaping)    | MELR (with shaping) |
> | --- | --- | --- |
> | FNO  | 10.3 | **1.9** |
> | TF-Net | 4.2 | **0.6** |
> | Galerkin Transformer | **9.4** | 10.1 |
> | Modern UNet | 6.5 | **0.8** |
> | PDE-Refiner (UNet) | 2.9 | - |
>
>
> Decorrelation time at 0.8 (first timestep when correlation drops below 0.8)
>
> | Model | Decorrelation time (no shaping)    | Decorrelation time (with shaping) |
> | --- | --- | --- |
> | FNO | 17.0 | **40.4** |
> | Galerkin Transformer | 13.5 | **16.5** |
> | TF-Net | 70.8 | **75.5** |
> | Modern UNet | 59.9 | **75.6** |
> | PDE-Refiner (UNet) | 76.3 | - |
>
> The findings we had in the paper that spectral shaping improves MELR holds for all baselines apart from the Galerkin Transformer, where, as we stated, it is not directly obvious how to apply spectral shaping. Furthermore, note how most models with spectral shaping outperform PDE-Refiner on MELR. Note that PDE-Refiner was originally developed and motivated with improving spectral shape in mind.
>
> In terms of decorrelation time, spectral shaping always improved this.
>
> **The method is only evaluated for a single model (U-Net). This leads to the question of whether the good experimental results are due to using a model that may be especially prone to producing high-frequency noise. For example, the paper could be improved by evaluating whether the filter helps FNO. This would also allow a comparison to the method of McCabe et al. (2023).**
>
> Please see above

---

> > ### Author Response · Authors · 2024-11-25
> > **Rebuttal (2/2)**
> >
> > **The right filter curve is found by hand and not learned.**
> >
> > If “by hand” you mean a grid-search, then this is true, just like many hyperparameters of practical interest, for instance the denoising variance schedule in PDE-Refiner. We find this aligns with standard machine learning practice. If you believe this to be a problem of our paper, it is true for most in machine learning.
> >
> > **Why did you choose to train the model in float64 precision?**
> >
> > We trained the models in the native precision of the data. For 1D problems the data was generated with an ETDRK4 solver, which operates in float64. For the 2D problems, the data came in float32.
> >
> > **What do the circles in the box-plot in Fig. 8 represent?**
> >
> > These are outliers, as per box plot convention (please see https://en.wikipedia.org/wiki/Box_plot)

---

> > > ### Comment · Reviewer_1CqM · 2024-11-27
> > >
> > > Thank you for providing further experiments. They resolve my concern regarding the applicability to other models.
> > >
> > > Just to clarify, is the base model used in PDERefiner in these new experiments identical to the "modern U-Net"?
> > >
> > > Could you please also elaborate on the difference between your work and the spectral filtering by McCabe et al. [5]?
> > >
> > > > Models such as FNO, Galerkin Transformer, and other neural operator models are learned maps directly from initial conditions to solution values at time t. Meanwhile the focus of this paper is on autoregressive methods. That said, nothing stops us from training a Neural Operator model in an autoregressive fashion and we do so for the rebuttals, but this is really not the arena where such models shine.
> > >
> > > While FNO, for example, was introduced for a direct prediction problem, it was used extensively as an autoregressive model, for example, in PDERefiner [3] or PDEBench [2]. The Markov neural operator [4], for example, was explicitly developed for an autoregressive setting.  Additionally, the U-Nets [1] used in this paper were also not originally proposed for an autoregressive task but rather for image segmentation.
> > >
> > > > If “by hand” you mean a grid-search, then this is true, just like many hyperparameters of practical interest, for instance the denoising variance schedule in PDE-Refiner. We find this aligns with standard machine learning practice. If you believe this to be a problem of our paper, it is true for most in machine learning.
> > >
> > > The point about the parameter is that it introduces an extra parameter on top of the ones already existing in the model/training procedure, thereby potentially increasing the hyperparameter tuning effort.
> > >
> > > ------
> > >
> > > [1] Ronneberger et al. (2015). U-net: Convolutional networks for biomedical image segmentation. In Medical image computing and computer-assisted intervention–MICCAI 2015: 18th international conference, Munich, Germany, October 5-9, 2015, proceedings, part III 18 (pp. 234-241). Springer International Publishing.
> > > [2] Takamoto et al. (2022). Pdebench: An extensive benchmark for scientific machine learning. Advances in Neural Information Processing Systems, 35, 1596-1611.
> > > [3] Lippe et al. (2024). Pde-refiner: Achieving accurate long rollouts with neural pde solvers. Advances in Neural Information Processing Systems, 36.
> > > [4] Li et al. (2022). Learning dissipative dynamics in chaotic systems. In Proceedings of the 36th International Conference on Neural Information Processing Systems (pp. 16768-16781).
> > > [5] McCabe et al. (2023). Towards stability of autoregressive neural operators. Transactions on Machine Learning Research..

---

> > > > ### Author Response · Authors · 2024-11-27
> > > > **Clarifications**
> > > >
> > > > Thanks for the acknowledgement. We are glad the experiments are convincing.
> > > >
> > > > **Just to clarify, is the base model used in PDERefiner in these new experiments identical to the "modern U-Net"?**
> > > >
> > > > Those numbers are with a regular UNet. We are currently training on a modern UNet for comparison (we will need this anyway for the paper), but it's proven slower to train and requires a lot of parameter tuning.
> > > >
> > > > **Could you please also elaborate on the difference between your work and the spectral filtering by McCabe et al. [5]?**
> > > >
> > > > We highlight 3 main differences:
> > > > 1. McCabe's paper focuses specifically on stabilizing Neural Operators (like FNO) through learned filtering mechanisms integrated into the Neural Operator architecture. While effective, this approach is specialized to that class of models. Our method of applying fixed Gaussian filtering after nonlinearities is a more general result that is agnostic to the underlying architecture (as long as there is a spatially-structured latent space), making it more broadly applicable across different types of neural PDE surrogates.
> > > >
> > > > 2. McCabe is more adaptive since the filter is learnable, but that also makes it architecture specific. One would have to come up with ways to implement it for every architecture you might use and it's harder to find efficient approaches when you're not operating directly in the spectral domain. Spectral Shaping on the other hand is less flexible, but can be applied out of the box to absolutely any architecture.
> > > >
> > > > 3. McCabe also requires a fixed input size to efficiently parameterize the learned filter, since Spectral Shaping is just a fixed filter, it can be applied anywhere.
> > > >
> > > > **While FNO, for example, was introduced for a direct prediction problem, it was used extensively as an autoregressive model**
> > > >
> > > > Agreed. The main issue we were bringing up is the typical use-case for such models, which have sometimes proven less well suited for autoregressive problems. Nonetheless, we included autoregressive FNO and Galerkin Transformer and our results still hold.
> > > >
> > > > **The point about the parameter is that it introduces an extra parameter on top of the ones already existing in the model/training procedure, thereby potentially increasing the hyperparameter tuning effort.**
> > > >
> > > > Yes, we have an extra parameter to tune. Here is one more perspective on why learning is very difficult, on top of the one we offered earlier to do with overfitting. The magnitude of high frequency signals is very small. FNO-type approaches, in general, require high numerical precision to reasonably learn good filters in the high frequency range, because those signals are so small. Furthermore, the error signal in the high end of the range is drowned out by error at lower frequencies, which are generally higher—the so-called spectral bias. The fixed filter is less adaptive, but also has well understood behavior across all scales without needing to directly learn from low magnitude data at the top end of the spectrum.

---

### Official Review · Reviewer_6BVv · 2024-11-03

**Soundness:** 2
**Presentation:** 3
**Contribution:** 2
**Rating:** 5
**Confidence:** 2

**Summary:**

The paper proposes a filtering-based approach to achieve long rollouts of neural PDE solvers. Neural surrogates for PDE solvers face issues in simulating problems for large temporal domains, and autoregressive-based methods, which this paper focuses on, have an additional problem with the accumulation of errors over time steps. The proposed method applies a low-pass filter after each pointwise nonlinearity in the network to mitigate the so-called spectral junk in long rollouts. The method is presented on popular PDE problems shown as challenging in the literature.

**Strengths:**

The proposed method is simple, easy to implement and is shown to be effective.

The proposed filtering step can be combined with different methods and can help, in general, the long rollout of PDE solvers.

The choice of PDEs is relevant to the problem being solved.

**Weaknesses:**

First, a general comment is that the paper is hard to read and assumes the reader is well-read with the literature. Further explanation of concepts, such as defining how the PDE solutions' low, medium, and high-frequency bands are identified and explaining the eigenvalue spectrum of the linearized flow map, would help the reader understand the challenges and motivate them to read the work.

The chosen baselines are vanilla, and more advanced variants of the architectures could have been considered as the baseline, for e.g. advanced variants of U-Net have been applied for temporal conditioning and long rollouts.

The challenge of the long rollout is even persistent for 3D stiff problems; how well the method scales to a higher-dimensional problem still needs to be checked.

The proposed method relies on filtering strategies, however, a detailed discussion of different filters, their pros, cons, and rationale for choosing the specific filtering approach is missing.

**Questions:**

One of the popular methods for long rollouts is PDE-Refiner. The authors discuss it briefly but do not compare it against the method. A comparison with PDE-Refiner and the pros and cons of both methods will help the reader understand how the method fairs against the methods established in the literature.

A detailed study regarding the performance of filtering steps needs to be carried out, such as: How sensitive is the performance to the choice of filtering parameters? Would an adaptive filtering mechanism be feasible? A study with different filters and their performance in the proposed setting would also aid in understanding the robustness of the proposed approach.

Did the authors experiment with placing the filter at different stages in the network (e.g., after every few layers instead of each nonlinearity), and how did that affect performance?

Since the authors argue that MSE loss biases the model towards low frequencies, How would the proposed method behave with alternative loss functions that could complement spectral shaping?

As a general remark, how does the proposed method guarantee that the errors are smoothed rather than smoothing out important features of the solution? For instance, how will the method perform for problems involving bifurcations?

How does the method perform regarding computational cost compared to the baselines?

---

> ### Author Response · Authors · 2024-11-25
> **Rebuttal (1/2)**
>
> Thanks for the review. We have tried to address all critique and suggestions to the best we can in the given time window. If you too believe we have addressed some of your concerns please do revise you score to reflect this.
>
> **First, a general comment is that the paper is hard to read and assumes the reader is well-read with the literature. Further explanation of concepts, such as defining how the PDE solutions' low, medium, and high-frequency bands are identified and explaining the eigenvalue spectrum of the linearized flow map, would help the reader understand the challenges and motivate them to read the work.**
>
> Thanks for this feedback. We have tried to strike a balance between explaining everything from scratch and keeping things short, to cater to a broad readership and indeed some reviewers had the opposite experience of the paper. We will update the explanations in the areas you mention to improve readability. Specifically the different frequency bands are more qualitative regions than anything very specific, read, left, middle and right of spectra, but we can tighten this up by referring to the “energy containing range”, “inertial range”, and “dissipative range” of classical kinetic turbulent energy spectra. Regarding the eigen-spectrum, we can provide an exact description in the Appendix.
>
> **The chosen baselines are vanilla, and more advanced variants of the architectures could have been considered as the baseline, for e.g. advanced variants of U-Net have been applied for temporal conditioning and long rollouts.**
>
> The original idea to focus on simple baselines was motivated by the fact that phenomena are easier to analyze. That said, we understand in such a deeply empirical field demonstrating an idea works is more convincing on a wider range of models. We accept this criticism and present a number of extra baselines. We provide these for the KS equation, which is all we had time for in the rebuttal period
>
> We have added some more baselines. But first an explanation why we did not initially include them. Models such as FNO, Galerkin Transformer, and other neural operator models are learned maps directly from initial conditions to solution values at time t, namely, $u(., 0) -> u(., t)$. Meanwhile the focus of this paper is on autoregressive methods. That said, nothing stops us from training a Neural Operator model in an autoregressive fashion and we do so for the rebuttals, but this is really not the arena where such models shine. Regarding Galerkin Transformer, notions of spatial geometry are lost in the attention layer and so it is not obvious how to apply spectral shaping in that space. PDE-Refiner, a diffusion-type model, is stochastic and was not the focus of this paper, but we include it, since it is state-of-the-art. It cannot be combined with spectral shaping.
>
> We provide these for the KS equation, which is all we had time for in the rebuttal period. We report MELR (Mean energy log ratio, a measure of spectral shape) at 128 time steps (lower is better) and decorrelation time  at 0.8 Pearson’s correlation coefficient (higher is better).
>
> (**Bold** indicates best in each row)
>
> | Model | MELR (no shaping)    | MELR (with shaping) |
> | --- | --- | --- |
> | FNO  | 10.3 | **1.9** |
> | TF-Net | 4.2 | **0.6** |
> | Galerkin Transformer | **9.4** | 10.1 |
> | Modern UNet | 6.5 | **0.8** |
> | PDE-Refiner (UNet) | 2.9 | - |
>
> Decorrelation time at 0.8 (first timestep when correlation drops below 0.8)
>
> | Model | Decorrelation time (no shaping)    | Decorrelation time (with shaping) |
> | --- | --- | --- |
> | FNO | 17.0 | **40.4** |
> | Galerkin Transformer | 13.5 | **16.5** |
> | TF-Net | 70.8 | **75.5** |
> | Modern UNet | 59.9 | **75.6** |
> | PDE-Refiner (UNet) | 76.3 | - |
>
> The findings we had in the paper that spectral shaping improves MELR holds for all baselines apart from the Galerkin Transformer, where, as we stated, it is not directly obvious how to apply spectral shaping. Furthermore, note how most models with spectral shaping outperform PDE-Refiner on MELR. Note that PDE-Refiner was originally developed and motivated with improving spectral shape in mind.
>
> In terms of decorrelation time, spectral shaping always improved this.
>
>
> **The challenge of the long rollout is even persistent for 3D stiff problems; how well the method scales to a higher-dimensional problem still needs to be checked.**
>
> This is certainly true. For papers targeting machine learning conferences, we believed that 3 equations in 1D and 2D would be enough. 3D problems present an entirely different beast, that many in the field do not come on to. We would like to keep it that way for this paper

---

> > ### Author Response · Authors · 2024-11-25
> > **Rebuttal (2/2)**
> >
> > **The proposed method relies on filtering strategies, however, a detailed discussion of different filters, their pros, cons, and rationale for choosing the specific filtering approach is missing.**
> >
> > We found between Gaussian, Chebyshev, and Butterworth filters and found them all to perform similarly. The Gaussian filter was simple enough, which is why we just focussed on that. We will add the ablations to the appendix, but you will find that the effects are small. Essentially, any filter which suppresses high frequencies works.
> >
> > **One of the popular methods for long rollouts is PDE-Refiner. The authors discuss it briefly but do not compare it against the method. A comparison with PDE-Refiner and the pros and cons of both methods will help the reader understand how the method fairs against the methods established in the literature.**
> >
> > As we state in the paper, we focus on deterministic methods. But we include new results comparing against PDE-Refiner above anyway. Spectral shaping is cheap and deterministic requiring just a single forward pass through the model; whereas, PDE-Refiner is expensive and stochastic, requiring multiple forward passes and noising injection. That said, it is a very nice model with good performance in terms of rollout decorrelation time. Note however that spectral shaping appears to have much better spectral shape than PDE-Refiner, which struggles to achieve MELR below 2.9 compared to spectral shaping at 0.8
> >
> > **A detailed study regarding the performance of filtering steps needs to be carried out, such as: How sensitive is the performance to the choice of filtering parameters? Would an adaptive filtering mechanism be feasible? A study with different filters and their performance in the proposed setting would also aid in understanding the robustness of the proposed approach.**
> >
> > Please see the above on ablations. We shall include this.
> >
> > **Did the authors experiment with placing the filter at different stages in the network (e.g., after every few layers instead of each nonlinearity), and how did that affect performance?**
> >
> > We only included filtering at the end versus after every activation function.
> >
> > **Since the authors argue that MSE loss biases the model towards low frequencies, How would the proposed method behave with alternative loss functions that could complement spectral shaping?**
> >
> > Please see “Learning to minimize spectral gap” in the paper, which does just this!
> >
> > **As a general remark, how does the proposed method guarantee that the errors are smoothed rather than smoothing out important features of the solution? For instance, how will the method perform for problems involving bifurcations?**
> >
> > This paper focuses on problems with smooth solutions. Solutions exhibiting shocks and bifurcations are not under the scope of this method.
> >
> > **How does the method perform regarding computational cost compared to the baselines?**
> >
> > In terms of wall-clock, we noticed no difference. The smoothing filters are extremely cheap to implement.

---

> > > ### Comment · Reviewer_6BVv · 2024-11-28
> > > **Response by reviewer**
> > >
> > > Thank you for answering the questions and further experiments. The reason I asked for the performance on further complicated problems and higher-dimensional problems, relate to the fact that the proposed method shows that a minor filter update step could lead to correct solutions in long rollouts. This claim is too strong to make given the experimentation presented. It could be possible that the method works fairly well in complicated problems as well, however, as the authors do not provide further experiments (and also cannot provide in this discussion time), I will continue with my initial assessment.

---

### Official Review · Reviewer_ru7e · 2024-11-04

**Soundness:** 2
**Presentation:** 3
**Contribution:** 3
**Rating:** 5
**Confidence:** 5

**Summary:**

This paper proposes a method to post-process the spectra of the latent fields as well as the output for the predictions from a neural surrogate model (applying a low-pass filter for every latent field after activation). This is definitely an interesting paper that tries to address a commonly recognized problem for the neural surrogates model (the combination of distribution shift and the spectral bias) from an architectural pov.

**Strengths:**

- This paper chose interesting datasets to study. All datasets are NS-based, i.e., there will always be an energy cascade forming after a certain "burn-in" time, so one has stable statistics to play with and to compare with.
- The proposed method is simple.

**Weaknesses:**

There is no theoretical analysis of why low-pass filter improves the stability of the repeated application of the flow map. The lack of this sort of analysis is okay though and does not affect my final score. There are many details for experiments or methods used in the experiments missing, examples:
- No modern baselines, such as FNO or FNO variants, Transformer-based NOs (Galerkin, OFormer, GNOT, and many others), or DeepONets.
- There should be an ablation of different filters (at least some filters with sharper cut-offs than Gaussian, Butterworth for example).
- The linearized flow map is unclear, is the linearization at the current time step?
- There are some minor notational errors, such as the PDE for the Kolmogorov flow is wrong, and the divergence-free condition in RBC is written as something else.
- I would argue differently from Figure 8 on the usage of only filtering the last layer vs every layer. RBC is a different beast than others, as one of the major difficulties for neural surrogates is the boundary conditions (not so big for periodic boxes) after repeated application.
- The training done in fp64 for 1D problem and fp32 for 2D problem.
- The authors wrote "the choice of $\sigma$ is important", but never reported the comparison.

**Questions:**

- Why not making $\sigma$ learnable in the filter?
- What is a "high-frequency regularizer"?

---

> ### Author Response · Authors · 2024-11-25
> **Rebuttal (1/2)**
>
> Thanks for your review. We are generally satisfied and believe that you have ingested the central ideas of the paper. Below we address your comments and critique. We hope this improves the paper. If you also believe this, please update your rating to reflect this.
>
> **No modern baselines, such as FNO or FNO variants, Transformer-based NOs (Galerkin, OFormer, GNOT, and many others), or DeepONets.**
>
> We have added some more baselines. But first an explanation why we did not initially include them. Models such as FNO, Galerkin Transformer, and other neural operator models are learned maps directly from initial conditions to solution values at time t, namely, $u(., 0) -> u(., t)$. Meanwhile the focus of this paper is on autoregressive methods. That said, nothing stops us from training a Neural Operator model in an autoregressive fashion and we do so for the rebuttals, but this is really not the arena where such models shine. Regarding Galerkin Transformer, notions of spatial geometry are lost in the attention layer and so it is not obvious how to apply spectral shaping in that space. PDE-Refiner, a diffusion-type model, is stochastic and was not the focus of this paper, but we include it, since it is state-of-the-art. It cannot be combined with spectral shaping.
>
> We provide these for the KS equation, which is all we had time for in the rebuttal period. We report MELR (Mean energy log ratio, a measure of spectral shape) at 128 time steps (lower is better) and decorrelation time  at 0.8 Pearson’s correlation coefficient (higher is better).
>
> (**Bold** indicates best in each row)
>
> | Model | MELR (no shaping)    | MELR (with shaping) |
> | --- | --- | --- |
> | FNO  | 10.3 | **1.9** |
> | TF-Net | 4.2 | **0.6** |
> | Galerkin Transformer | **9.4** | 10.1 |
> | Modern UNet | 6.5 | **0.8** |
> | PDE-Refiner (UNet) | 2.9 | - |
>
>
> Decorrelation time at 0.8 (first timestep when correlation drops below 0.8)
>
> | Model | Decorrelation time (no shaping)    | Decorrelation time (with shaping) |
> | --- | --- | --- |
> | FNO | 17.0 | **40.4** |
> | Galerkin Transformer | 13.5 | **16.5** |
> | TF-Net | 70.8 | **75.5** |
> | Modern UNet | 59.9 | **75.6** |
> | PDE-Refiner (UNet) | 76.3 | - |
>
>
> The findings we had in the paper that spectral shaping improves MELR holds for all baselines apart from the Galerkin Transformer, where, as we stated, it is not directly obvious how to apply spectral shaping. Furthermore, note how most models with spectral shaping outperform PDE-Refiner on MELR. Note that PDE-Refiner was originally developed and motivated with improving spectral shape in mind.
>
> In terms of decorrelation time, spectral shaping always improved this.
>
> **There should be an ablation of different filters (at least some filters with sharper cut-offs than Gaussian, Butterworth for example).**
>
> We found between Gaussian, Chebyshev, and Butterworth filters and found them all to perform similarly. The Gaussian filter was simple enough, which is why we just focussed on that. We will add the ablations to the appendix, but you will find that the effects are small. Essentially, any filter which suppresses high frequencies works.
>
> **The linearized flow map is unclear, is the linearization at the current time step?**
>
> We assume you are referring to Figure 1a. Indeed, the linearization is performed about the current timestep for a randomly sampled trajectory in the test set. This is mainly intended as an exemplar of a phenomenon we have observed across many trajectories and timesteps.
>
> **There are some minor notational errors, such as the PDE for the Kolmogorov flow is wrong, and the divergence-free condition in RBC is written as something else.**
>
> Thanks for catching the error in the forcing for Kolmogorov flow! Regarding RBC, we have used the problem description as implemented in https://dedalus-project.readthedocs.io/en/latest/pages/examples/ivp_2d_rayleigh_benard.html
>
> **I would argue differently from Figure 8 on the usage of only filtering the last layer vs every layer. RBC is a different beast than others, as one of the major difficulties for neural surrogates is the boundary conditions (not so big for periodic boxes) after repeated application.**
>
> This is an interesting point. We do agree that for RBC the boundary conditions are important. However, what Figure 8 highlights is that the model is not able to achieve a lengthened decorrelation window without some level of filtering (or equivalent such as denoising, pushforward, etc.). Indeed, as you say, solving the boundary conditions is an important component in solving RBC, but it is complementary to spectral shaping, not in competition.

---

> ### Author Response · Authors · 2024-11-25
> **Rebuttal (2/2)**
>
> **The training done in fp64 for 1D problem and fp32 for 2D problem.**
>
> We chose the floating point format to match the format in which the data is generated. It did not appear to be much of an issue for us. Importantly, however, running the 1D problem in fp64 highlights how spectral shaping is able to perform with a much higher dynamic range compared to other method, such as PDE-Refiner.
>
> **The authors wrote "the choice of sigma is important", but never reported the comparison.**
>
> Please revisit Figure 8. You will notice the box plots show the range of results after sweeping over sigma. We will update the paper to highlight this fact.
>
> **Why not making sigma learnable in the filter?**
>
> We did try this and it did not work for us. We are not sure why, but the most likely answer is that sigma controls the ability of the model to overfit over a single step. Therefore, making sigma learnable will not work unless we add a regularizer on the sigma to encourage it towards larger values.
>
> **What is a "high-frequency regularizer"?**
> In “Learning to minimize spectral gap” we added the MELR loss to the L2 loss. This is in effect a regularizer on the high frequency content to match the ground truth. Alone this is not enough to encourage the model to match spectra, low pass filtering is needed. That is what we meant. We shall update the paper accordingly.

---

> > ### Comment · Reviewer_ru7e · 2024-11-27
> >
> > The pdf is not updated. Is this a revision I should check somewhere?

---

> > > ### Author Response · Authors · 2024-11-27
> > >
> > > The deadline for updates to PDFs has officially passed. Nonetheless, l the updated numbers, extra theory, and replies to your comments above will be factored into a our latest version.

---

> > > > ### Comment · Reviewer_ru7e · 2024-11-30
> > > >
> > > > Thanks for the response. I will keep my score due to no revision being uploaded.
> > > >
> > > > BTW: the pdf update deadline was Nov 28 6am my local time (Nov 27 AoE, about 15 hours from the time stamp your comment was posted).

---

### Meta-Review · Area_Chair_ZQB4 · 2024-12-20

**Metareview:**

This paper proposes a spectral post-processing (filtering) method for neural PDE surrogates in an autoregressive, time-stepping mode. This is shown to stabilize long rollouts. The reviewers all thought that the paper is interesting and that it addresses a real, important problem. But these positive aspects were outweighted by critical remarks about the writing, hardness of experiments, and an agreement that the conclusions are not supported by available theory and experiments. Dealing with these aspects would make this a fine paper, good for publication at one of the upcoming ML venues.

**Additional Comments On Reviewer Discussion:**

6BVv convincingly argued that the claims made in the paper are not supported by the provided experiments; in particular, the one that "minor filter update step could lead to correct solutions in long rollouts." 1CqM also pointed out weak baselines and asked why there are no comparisons with strong baselines like FNOs. The authors responded at the very end of the initial discussion period. While they did include some additional experiments there was a consensus that the paper is still not ready for publication.
Overall, the authors' responses were often unconvincing.
About baselines, the authors argued that FNOs are not meant to be used in autoregressive schemes. 1CqM pointed out a number of strong papers which use FNOs exactly in this way. Finally, the authors did not upload a revised pdf...

I base my recommendation on a consesus among the reviews that the paper, with the current organization, narrative, and experiments, is not ready for publication, as well as on a careful reading of the discussion. I'm sure that it could be a very solid paper with some work.

---

### Decision · Program_Chairs · 2025-01-22

Reject